# Attention Sinks in Diffusion Transformers:
# A Causal Analysis

**Fangzheng Wu** [1]  **Brian Summa** [1]

## Abstract

Attention sinks—tokens that receive disproportionate attention mass—are assumed to be functionally important in autoregressive language models, but their role in diffusion transformers remains unclear. We present a causal analysis in text-to-image diffusion, dynamically identifying dominant attention recipients per timestep and suppressing them via paired, training-free interventions on the score and value paths. Across 553 GenEval prompts on Stable Diffusion 3 (with SDXL corroboration), removing these sinks does not degrade text-image alignment (CLIP-T) or preference proxies (ImageReward, HPS-v2) at $k=1$; only under stronger interventions ($k \geq 10$) does HPS-v2 exhibit a metric-dependent boundary, while CLIP-T remains robust throughout. The perceptual shifts induced by suppression are nonetheless *sink-specific*—$\sim 6\times$ larger than equal-budget random masking—revealing an empirical dissociation between trajectory-level perturbation and *semantic alignment* in diffusion transformers. [1]

## 1. Introduction

Attention mechanisms in large language models frequently converge on specific *attention sinks*—tokens or positions that accumulate disproportionate attention mass independently of their semantic importance (Xiao et al., 2024; Sun et al., 2024). In autoregressive language models, such sinks have been interpreted as functionally important structures: they serve as stable anchors for key-value caching in long-context generation, prevent attention entropy collapse, and

act as implicit registers for residual information. Empirically, removing or disrupting sinks in autoregressive models can degrade generation quality or destabilize inference. As a result, attention sinks are often treated not as statistical artifacts but as load-bearing computational structures in transformer-based generation.

However, it is unclear whether this intuition transfers to diffusion transformers. Unlike autoregressive decoding, which proceeds token-by-token under causal masking, diffusion models perform non-causal, bidirectional attention across multiple denoising steps, iteratively refining all positions simultaneously rather than committing to irreversible left-to-right decisions (Peebles & Xie, 2023; Saharia et al., 2022). In this setting, attention does not obviously serve as a persistent memory anchor for preceding context (a role often attributed to sinks in autoregressive settings). This architectural distinction raises a natural question: *are attention sinks equally necessary under diffusion-style inference, or does the intuition from autoregressive models fail to transfer in practice?* The answer matters concretely for efficient diffusion attention design, especially in DiT-style architectures. Recent sparsification and attention-compression methods make different implicit assumptions about high-mass attention recipients: some preserve tokens through attention-driven importance scores (Wang et al., 2024), whereas others impose locality or structured sparsity patterns (Yuan et al., 2024; Ren et al., 2025; Chang et al., 2026) that may discard such recipients depending on the layer and timestep. This leaves open a basic but consequential ambiguity: does high incoming attention mass indicate functional necessity, or merely mark a replaceable routing pattern? Without causal evidence on sink necessity, these design choices lack grounding for which high-mass recipients to preserve.

Existing reports of sink-like phenomena in diffusion-style models (Arriola et al., 2025; Wen et al., 2025; Rulli et al., 2025) suggest some dominant recipients are removable without catastrophic failure, but the evidence is fragmented—definitions vary across studies (fixed-position vs. dynamic), interventions are often observational rather than causal, and evaluations are limited to specific modalities or small-scale settings. Whether attention sinks are truly *necessary* for high-quality generation in text-to-image diffusion transform-

---

[1]Department of Computer Science, Tulane University, New Orleans, LA, USA. Correspondence to: Fangzheng Wu <fwu6@tulane.edu, fwu66666666@gmail.com>.

*Proceedings of the $43^{rd}$ International Conference on Machine Learning*, Seoul, South Korea. PMLR 306, 2026. Copyright 2026 by the author(s).

[1]Code available at https://github.com/wfz666/ICML26-attention-sink.

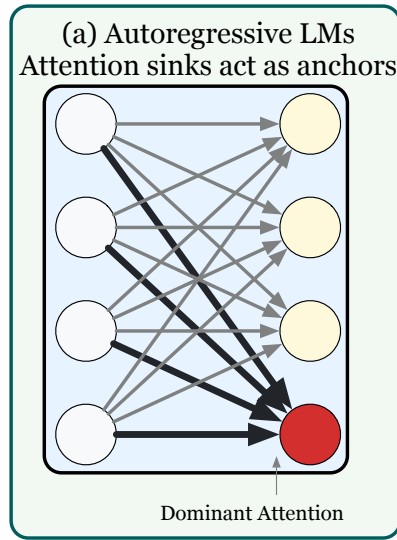 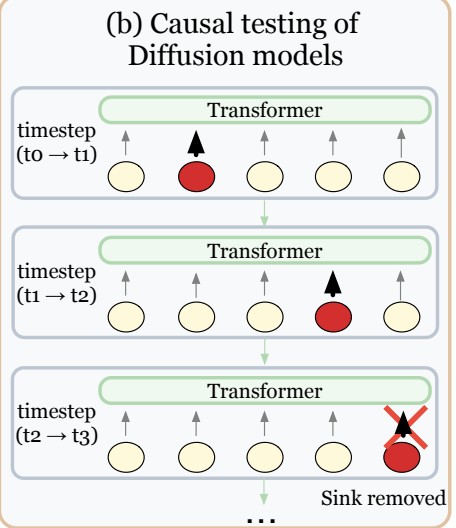 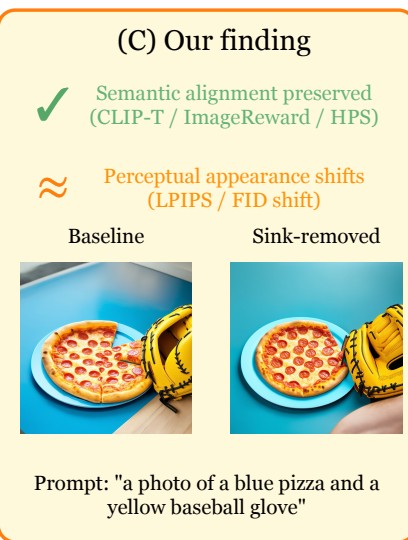

*Figure 1.* **Attention sinks in diffusion transformers.** **(a)** In autoregressive LMs, attention sinks often act as stable anchors that attract dominant attention mass. **(b)** In diffusion transformers, dominant recipients vary across denoising timesteps; we perform a *causal* test by dynamically identifying sink tokens per step and suppressing them during inference. **(c)** Sink suppression preserves semantic alignment and preference scores (CLIP-T / ImageReward / HPS-v2), yet can induce perceptual and distributional shifts relative to baseline outputs (LPIPS / FID$_{shift}$), consistent with moving samples within the model's output manifold.

ers thus remains unresolved.

In this work, we provide a systematic causal analysis of attention sinks in diffusion transformers (DiT) (Peebles & Xie, 2023). We define sinks dynamically as key positions receiving the largest incoming attention mass, separately for each attention head and denoising timestep. This approach moves beyond the fixed-position assumptions typical of autoregressive settings, which our empirical data suggest are largely invalid (index-0 overlap <0.2%). Using paired, training-free interventions along both the score (logit) and value paths, we test the necessity of sinks across layers, denoising phases, intervention intensities, and architectures (SD3 (Esser et al., 2024) and SDXL (Podell et al., 2024)). All experiments employ strict seed-matched generation with bootstrap confidence intervals, ensuring that observed differences reflect intervention effects rather than sampling variance.

Across large-scale evaluations on 553 prompts from GenEval (Ghosh et al., 2023) with Stable Diffusion 3, and corroborating experiments on 5,000 COCO captions and SDXL, we find that removing dynamically identified attention sinks does not degrade semantic alignment (CLIP-T (Hessel et al., 2021)) or preference metrics (ImageReward (Xu et al., 2023), HPS-v2 (Wu et al., 2023)), which are trained on large-scale human preference datasets such as Pick-a-Pic (Kirstain et al., 2023). At the same time, suppressing dominant recipients induces perceptual shifts that are *sink-specific*—roughly $6\times$ larger than equal-budget random masking—revealing an empirical dissociation between trajectory-level perturbation and alignment-level robustness

(Figure 1). Together, these findings clarify that attention sinks are not functionally necessary for semantic alignment in diffusion transformers, while identifying sink-specific perceptual drift as a structural boundary condition of sink suppression.

More broadly, our results provide causal evidence that incoming attention mass—widely used as a proxy for token importance—does not reliably predict functional necessity for semantic alignment in diffusion transformers. We focus on alignment necessity at the level of widely used proxy metrics (CLIP-T, ImageReward, HPS-v2), treating perceptual changes as boundary conditions and additionally characterizing a metric-dependent boundary under stronger interventions where preference proxies show sink-specific effects while alignment remains stable. Skill-based and holistic evaluation frameworks—such as Gecko (Wiles et al., 2025) and HEIM (Lee et al., 2023)—that probe finer compositional fidelity remain a natural complement for future work.

**Conflict of Interest Disclosure.** The authors have no financial or other substantive conflicts of interest to disclose.

## 2. Related Work

**Attention sinks in autoregressive language models.** In autoregressive (AR) language models, certain tokens—often the beginning-of-sequence token or early positions—receive disproportionately high attention mass regardless of their semantic relevance (Xiao et al., 2024; Sun et al., 2024). These "attention sinks" have been linked to several func-

tional roles: serving as stable anchors for key-value caching in long-context generation (Xiao et al., 2024), preventing attention entropy collapse (Sun et al., 2024), and acting as implicit registers that accumulate residual information. Similar "register" phenomena have been observed in discriminative vision transformers (Darcet et al., 2024), raising the question of whether such structures serve analogous functions in generative diffusion transformers. Jiang et al. (2026) subsequently show that the high-norm activations responsible for register-like behavior can be relocated at inference into an additional untrained token, achieving the effect of trained registers without retraining; this suggests register-style structures may be less load-bearing than initially thought even in discriminative settings, complementing our finding of sink dispensability in the generative diffusion regime. Empirically, removing or disrupting sinks in AR models can degrade generation quality or destabilize long-context performance (Gu et al., 2025). These findings shaped a widespread intuition that attention sinks are functionally necessary structures. **A common but largely untested assumption is that this functional role transfers to diffusion-style bidirectional denoising.**

**Sink-like phenomena in diffusion-based models.** Recent work has begun to probe attention concentration in non-autoregressive settings, with results that diverge from the AR narrative. In discrete diffusion language models, Arriola et al. (2025) observe sink-like attention patterns and report that certain dominant recipients can be ablated without catastrophic degradation, suggesting a potential commonality across discrete and continuous diffusion regimes, though systematic causal verification remains limited. Concurrent work by Rulli et al. (2025) reports strikingly parallel findings in masked diffusion language models: sinks exhibit dynamic positions throughout generation and remain largely robust to suppression, providing independent convergent evidence in the discrete-token regime that sink-like phenomena recur across diffusion architectures. For video diffusion transformers, Wen et al. (2025) identify redundant attention heads and demonstrate that certain connections can be pruned, with caveats regarding layer sensitivity. Critically, definitions of sinks vary across these studies: AR models exhibit *positionally anchored* sinks tied to structural tokens (e.g., BOS), whereas diffusion models may exhibit *dynamically varying* concentration that shifts across timesteps. Our finding that index-0 overlap is negligible (<0.2%, often <0.1% in our measurements) confirms this distinction and underscores the need for dynamic sink definitions in diffusion settings.

**Attention sparsification and acceleration.** Recent work explores attention compression for diffusion transformers, including window attention (Yuan et al., 2024), token pruning (Wang et al., 2024), linear attention (Xie et al., 2024),

and phase-aware caching (Zhao et al., 2025; Liu et al., 2024). These methods demonstrate that sparsity can be beneficial, *yet they do not isolate whether dominant recipients are causally necessary*. Our work provides the missing causal evidence, complementing recent structured sparsification approaches such as GRAT (Ren et al., 2025). Linear- and gated-linear-attention diffusion variants (Meng et al., 2026; Zhu et al., 2025) modify the attention operator itself rather than the recipients within softmax attention. Our conclusions are scoped to standard softmax-attention diffusion architectures; we do not claim the same necessity profile holds under alternative attention mechanisms (e.g., linear attention).

**Causal diagnostics at the head level.** Complementary to token-level analysis, prior work has studied which *attention heads* matter for generation. Michel et al. (2019) show that many heads can be pruned with minimal quality loss in language models. In diffusion settings, head-level gating has been explored for efficiency (Li et al., 2023b). These approaches ask "which heads matter"; we pursue an *orthogonal* inquiry: "do the most attended tokens within active heads matter?" Our results suggest that dominant attention recipients are non-functional even in heads that remain active, suggesting that head-level and token-level redundancy may be complementary and warrant separate investigation.

**Attention editing and modulation in diffusion models.** A separate line of work manipulates attention patterns to improve text–image alignment and compositional fidelity (Tang et al., 2023). Attend-and-Excite (Chefer et al., 2023) amplifies cross-attention to neglected tokens during inference, improving object presence. Earlier attention editing frameworks such as Prompt-to-Prompt (Hertz et al., 2022) demonstrate that selectively manipulating cross-attention can steer semantic content without retraining. More recent attention regulation methods explicitly reweight dominant attention patterns to improve compositional fidelity (Zhang et al., 2024). Structured cross-attention methods (Feng et al., 2022) decompose prompts to reduce attribute binding errors. More recently, phase-aware attention modulation has been explored for layout control (Chen et al., 2024). These methods demonstrate that *modifying* dominant attention can improve alignment; our work addresses the complementary question of whether such dominant recipients are *necessary* for alignment. Our results suggest they are not: suppressing high-mass recipients preserves alignment metrics under standard settings, indicating a clear distinction between *usefulness for optimization or controllability* and *necessity for semantic fidelity*.

# 3. Experiments and Results

## 3.1. Experimental Setup

**Models and evaluation.** We use SD3 (joint attention over image/text tokens) as our primary testbed and validate on SDXL (U-Net cross-attention); all models are used in inference mode without finetuning. We evaluate on 553 GenEval prompts using strict seed-paired generation, reporting mean paired differences with 95% bootstrap CIs (1k resamples). Supplementary analyses use smaller prompt subsets; details are in Appendix C.

**Metrics.** We use CLIP-T (Hessel et al., 2021) as the primary alignment metric, with ImageReward and HPS-v2 as preference proxies. Perceptual shifts are quantified via LPIPS and FID$_{shift}$ (distributional distance between baseline and intervention outputs, rather than against real images, as a sample-domain measure of distributional drift caused by the intervention.). We adopt $|\Delta| < 0.002$ for CLIP-T as the practical equivalence margin, chosen relative to the empirical noise floor observed under seed variation and bootstrap uncertainty at $N{=}553$; further justification is in Appendix C.

**Interventions.** We suppress sinks via two pathways: (1) **score-path**: add $\log \eta$ to sink logits ($\eta{=}0$ effectively zeros attention); (2) **value-path**: replace sink value vectors with alternatives (zero, mean, or interpolation). Both are training-free and applied during inference. Consistent conclusions across both pathways suggest robustness beyond any single masking scheme.

We emphasize that our claims concern *inference-time aggregation necessity*—whether suppressing sink tokens during the forward pass degrades the evaluated metrics—and are distinct from training-time functional roles such as gradient stabilization, which we do not test.

## 3.2. H1: Dynamics of Attention Sinks

We first characterize the *existence*, *layer-wise distribution*, and *temporal dynamics* of attention sinks during diffusion sampling.

**Dynamic Sink Definition.** Rather than assuming a fixed sink position (e.g., index-0 as in autoregressive LLMs), we define attention sinks dynamically based on incoming attention mass. For each attention head $h$ at layer $\ell$ and timestep $t$, we compute the *incoming attention mass* for each key position $j$:

$$m_j^{(\ell,t,h)} = \frac{1}{N} \sum_{i=1}^{N} A_{i,j}^{(\ell,t,h)}, \tag{1}$$

*Table 1.* **Attention concentration statistics (GenEval, $N = 553$).** $M_{mass}$: Max attention mass; $T_5$: Top-5 concentration; $A_{idx}$: Max activation index; $O_{idx0}$: Overlap with index-0. Middle layers exhibit peak concentration while remaining distinct from index-0.

| Layer | $M_{mass}$ | Entropy | $T_5$ Conc. | $A_{idx}$ | $O_{idx0}$ |
|---|---|---|---|---|---|
| 6 (shallow) | 1.9% | 6.5–7.0 | $0.11 \rightarrow 0.24$ | 7–10 | <0.2% |
| **12 (middle)** | **9.5%** | **4.0–4.5** | **$0.31 \rightarrow 0.42$** | **15–21** | **<0.2%** |
| 18 (deep) | 5.1% | 5.0–6.5 | $0.19 \rightarrow 0.29$ | 9–13 | <0.1% |

where $A_{i,j}^{(\ell,t,h)}$ denotes the attention weight from query $i$ to key $j$. The *dynamic top-$k$ sinks* are then defined as:

$$S^{(\ell,t,h)} = \mathrm{TopK}\big(m^{(\ell,t,h)}, k\big). \tag{2}$$

This definition identifies dominant attention recipients on a per-head, per-timestep basis without assuming any fixed token position. We note that $k$ is defined per head and per timestep; however, the effective number of unique masked tokens is substantially smaller than $H \times k \times T$, as multiple heads frequently select the same dominant text tokens across the sequence. We quantify sink strength using the *maximum incoming mass*:

$$\mathrm{MaxMass}^{(\ell,t)} = \frac{1}{H} \sum_{h=1}^{H} \max_j m_j^{(\ell,t,h)}. \tag{3}$$

Table 2 reports CLIP-T results for three conditions: single-layer (L12), multi-layer (L6+12+18), and stronger intervention (top-5). For the single-layer and multi-layer conditions, all 95% CIs include zero, confirming that removing dynamically identified sinks does not degrade quality.

The stronger intervention (top-5) yields a small positive shift in CLIP-T ($\Delta = +0.0011$, $p = 0.01$); this effect remains within our predefined practical equivalence margin ($|\Delta| < 0.002$). We evaluate this condition further with HPS-v2 below.

Together, Table 1 and Figure 2 yield five findings:

1. **Layer-wise concentration**: Layer 12 (middle) exhibits the strongest attention concentration, with maximum incoming mass $\approx 10\%$, lowest entropy (4.0–4.5), and highest activation magnitudes (15–21).
2. **Phase-dependent dynamics**: Following the timestep ordering of the official pipeline, we define normalized time $t/T \in [0, 1]$ such that $t/T \approx 0$ corresponds to the noisiest denoising step. Attention concentration peaks during early denoising and diminishes toward later steps.
3. **Concentration–entropy anti-correlation**: Layers with stronger attention concentration exhibit lower entropy, indicating more peaked attention distributions.
4. **Dynamic sink $\neq$ index-0**: Unlike autoregressive LLMs where attention sinks typically coincide with special tokens (e.g., BOS), the dynamically identified dominant

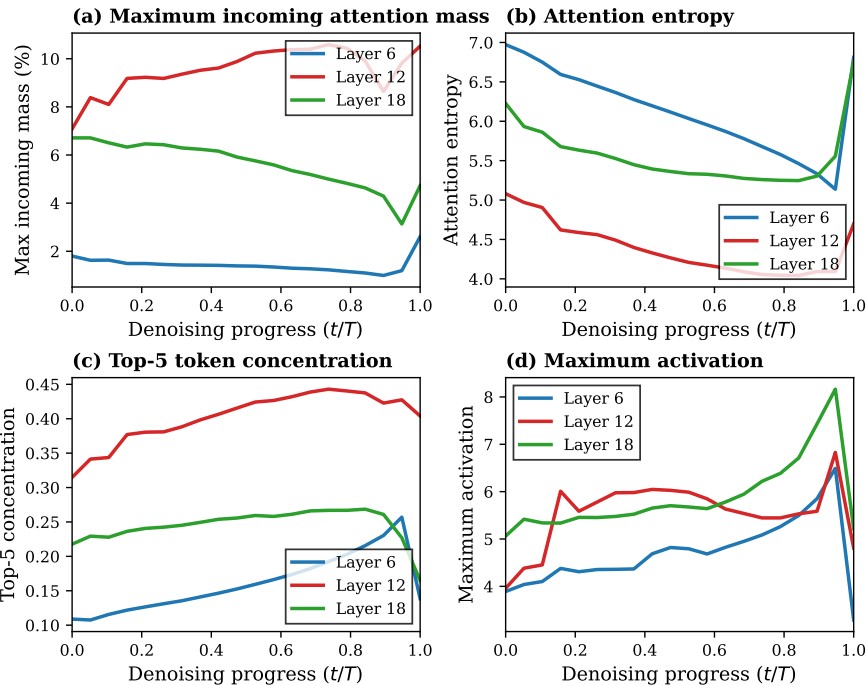

*Figure 2.* **Attention sink dynamics across layers and denoising timesteps.** (a) Maximum incoming mass peaks at Layer 12 and decreases during denoising. (b) Attention entropy is inversely correlated with attention concentration (MaxMass). (c) Top-5 concentration increases over time. (d) Maximum activation follows a similar pattern to MaxMass.

attention recipients in SD3 are *not* at index-0. The overlap between dynamic top-1 sinks and index-0 is $< 0.1\%$ across all layers and timesteps.

5. **Position locality**: Top-1 dynamic sinks consistently occupy a narrow range of key indices ($\approx$4016–4231 in SD3's joint sequence) corresponding predominantly to early text-encoder positions. The position drift across (layer, $t/T$) is structural rather than free, suggesting that "dynamic" refers to localized shifts within a small subset of conditioning positions rather than movement across the full key space. A token-identity decomposition of which specific text tokens these sinks correspond to (e.g., padding, structural tokens, or content-bearing tokens), in the spirit of the register-token analysis of Darcet et al. (2024) and Jiang et al. (2026), is left to future work.

These observations suggest that **attention concentration and dominant recipients** are transient and phase-dependent, emerging most prominently during the high-noise regime of early denoising.

### 3.3. H2: Causal Necessity of Attention Sinks

Having established that sinks exist, we now test whether they are *functionally necessary* for generation quality via causal interventions.

**Methodological note on "causal."** We use "causal" in the interventionist sense (Pearl, 2009): paired, training-free manipulations that directly alter internal computations while holding all other factors (prompt, seed, hyperparameters) fixed. This design isolates the effect of sink removal from confounding variation. While such ablations do not establish necessity for *all possible* downstream capabilities, they are sufficient to falsify necessity claims for the evaluated outcomes (alignment and preference metrics).

#### 3.3.1. DYNAMIC SINK INTERVENTIONS

We conduct causal interventions using our dynamic sink definition, which identifies the top-$k$ positions with highest incoming attention mass on a per-head, per-timestep basis.

For each attention head at the target layer(s), we compute the incoming mass distribution and identify the top-$k$ positions dynamically. We then apply a hard mask by subtracting $10^4$ from the pre-softmax logits of these positions, effectively zeroing their attention weights.

Table 2 reports results for three conditions: single-layer (L12), multi-layer (L6+12+18), and stronger intervention (top-5).

**Evaluation with a stronger preference model.** To mitigate concerns that CLIP-based metrics may miss subtle degradations, we additionally evaluate dynamic sink inter-

*Table 2.* **Dynamic sink intervention results (GenEval, $N = 553$).** All metric changes ($\Delta_C$) and 95% CIs are scaled by $10^3$. A value of 1.0 corresponds to a 0.001 absolute change in CLIP-T score. All primary shifts are negligible ($|\Delta_C| < 2.0$).

| Condition | Layers | $k$ | $\Delta_C$ ($\times 10^{-3}$) [95% CI] | $p$ |
|---|---|---|---|---|
| Single-L | 12 | 1 | $-0.5$ [$-1.7, +0.7$] | 0.44 |
| Multi-L | 6, 12, 18 | 1 | $+0.9$ [$-0.1, +2.0$] | 0.08 |
| Top-5 | 12 | 5 | $+1.1$ [$+0.3, +2.0$] | 0.01 |

*Table 3.* **HPS-v2 evaluation of dynamic sink interventions (GenEval, $N = 553$).** A1 and A2 show no significant change; A3 exhibits a small negative shift.

| Condition | Layers | $k$ | $\Delta$HPS-v2 | $p$ |
|---|---|---|---|---|
| A1 (Single-L) | 12 | 1 | $+0.0003$ | 0.42 |
| A2 (Multi-L) | 6, 12, 18 | 1 | $+0.0001$ | 0.85 |
| A3 (Top-5) | 12 | 5 | $-0.0020$ | 0.001 |

ventions using HPS-v2, a learned preference model trained on human preference data (Table 3).

For the primary conditions (A1: single-layer top-1; A2: multi-layer top-1), HPS-v2 shows no statistically significant change, with mean shifts near zero and 95% confidence intervals covering zero, consistent with CLIP-T and ImageReward.

Under the stronger A3 intervention (top-5 removal), HPS-v2 exhibits a small negative shift ($\Delta = -0.0020$, $p = 0.001$), while CLIP-T shows a small positive shift ($\Delta = +0.0011$). Importantly, both effects are below 1% in magnitude and lie near the boundary of practical equivalence, indicating a marginal metric-dependent trade-off rather than a meaningful quality change.

The intervention successfully eliminates attention to dynamically identified sinks. At Layer 12, the top-1 incoming mass is reduced from 10.0% to <0.001%, achieving reduction factors exceeding $10^8\times$ across all tested layers (Table 4).

These results significantly strengthen our main conclusion: even when sinks are identified dynamically as the true dominant attention recipients (rather than a fixed proxy position), their removal produces no measurable quality degradation.

Our interventions test *aggregation-level* necessity—whether sink tokens must contribute their value vectors for the evaluated outcomes. We do not test *encoding-level* necessity (whether sinks must contribute to key/query projections); doing so would require modifying upstream layers and is outside our scope.

*Table 4.* **Dynamic sink intervention verification (GenEval).** Reduction factors confirm effective removal of dominant attention recipients.

| Layer | Before | After | Reduction |
|---|---|---|---|
| 6 | 1.90% | <0.001% | $1.9 \times 10^8$ |
| **12** | **9.47%** | **<0.001%** | $\mathbf{9.5 \times 10^8}$ |
| 18 | 5.15% | <0.001% | $5.1 \times 10^8$ |

*Table 5.* **SDXL cross-architecture validation ($N = 100$, paired).** $\Delta$CLIP-T is the paired change in CLIP-T. Both modules yield 95% CIs containing zero; cross-attention induces larger perceptual drift, consistent with the SD3 attribution result that text-side sinks dominate trajectory perturbation (Section 3.6).

| SDXL Mid-Block | $\Delta$CLIP-T [95% CI] | LPIPS |
|---|---|---|
| Self-attention (`attn1`) | $+0.0004$ [$-0.0005, +0.0014$] | 0.045 |
| Cross-attention (`attn2`) | $-0.0003$ [$-0.0019, +0.0012$] | 0.077 |

### 3.4. Dose–Response Analysis

To rule out threshold effects, we perform full dose–response sweeps across both score-path ($\eta \in [0, 1]$) and value-path interventions. Both sweeps yield uniformly flat curves with all 95% CIs covering zero, even at complete removal ($\eta = 0$, reduction $> 44,000\times$). Full results are in Appendix D.

### 3.5. Robustness and Generalization

We test robustness across intervention scope, timing, and architecture. Full results are in Appendix E.

Simultaneous intervention on layers 6, 12, and 18 produces no degradation ($\Delta$CLIP-T $= +0.0012$, $p = 0.68$). Phase-specific interventions (early/mid/late only) likewise show no quality loss; notably, early-phase intervention when sink strength is maximal produces a slight *improvement* ($\Delta = +0.0006$), contradicting the hypothesis that sinks serve a critical anchoring function.

#### 3.5.1. CROSS-ARCHITECTURE VALIDATION: SDXL

To test generalization beyond SD3's joint attention, we intervene on both mid-block attention modules of SDXL: *self-attention* (`attn1`, image–image) and *cross-attention* (`attn2`, image–text, where keys/values are text embeddings). We focus on the mid-block as it concentrates the strongest text–image coupling; extending to early or late blocks is left for future work. Sanity checks confirm: (1) no-op produces pixel-identical outputs, and (2) intervention reduces sink mass by $> 10^9\times$.

Both modules yield CIs containing zero. Cross-attention induces larger perceptual drift (LPIPS 0.077 vs. 0.045), consistent with our SD3 E3 observation that text-side sinks produce larger trajectory perturbations (Section 3.6).

*Table 6.* **Perceptual and distributional effects.** LPIPS measures paired perceptual distance; $FID_{shift}$ measures distributional change between intervention and baseline outputs. Semantic alignment (CLIP-T) is preserved while perceptual similarity decreases with intervention intensity.

| Dataset | Cond. | $N$ | $\Delta$CLIP-T | LPIPS | $FID_{shift}$ |
|---------|-------|-----|----------------|-------|---------------|
| | A1 | 553 | +0.001 | 0.189 | 432 |
| GenEval | A2 | 553 | +0.001 | 0.242 | 992 |
| | A3 | 553 | −0.001 | 0.314 | 996 |
| COCO-5k | A1 | 5000 | −0.000 | 0.189 | 727 |
| *No-op* | – | 100 | – | 0.000 | 0 |

*Table 7.* **Sink vs. random masking at equal budget (union-budget protocol, layer 12,** $N$=64**).** $\Delta\Delta$ = $LPIPS_{sink}$ − $LPIPS_{rand}$. Sink masking induces substantially larger perceptual drift than equal-budget random masking at both budgets.

| $k$ | $LPIPS_{sink}$ | $LPIPS_{rand}$ | $\Delta\Delta$ | 95% CI | $p$ |
|-----|----------------|----------------|----------------|--------|-----|
| 1 | 0.347 | 0.053 | +0.295 | [+0.265, +0.323] | < 0.0001 |
| 5 | 0.436 | 0.104 | +0.332 | [+0.308, +0.358] | < 0.0001 |

### 3.5.2. PERCEPTUAL AND DISTRIBUTIONAL EFFECTS

Beyond alignment and preference metrics, we quantify how sink removal changes generated images relative to baseline outputs using paired perceptual distance (LPIPS) and same-domain distributional distance ($FID_{shift}$ computed between intervention and baseline outputs, not against real images).

Table 6 reports results across conditions. Perceptual and distributional shifts increase monotonically with intervention intensity: A1 (single-layer, top-1) induces moderate change (LPIPS $\approx 0.19$), while A3 (top-5) produces larger shifts (LPIPS $\approx 0.31$). Importantly, these shifts occur *without* degrading CLIP-T, ImageReward, or HPS-v2, indicating that suppressing dominant attention recipients moves samples within the model's output manifold while preserving standard alignment and preference scores.

To confirm that observed shifts arise from the intervention itself rather than implementation artifacts, we conduct a no-op sanity check: installing our attention processor with `intervention_enabled=False`. Across 100 paired generations, outputs are pixel-identical to unmodified baseline (pixel diff = 0, LPIPS = 0, $FID_{shift}$ = 0), confirming that the substantial perceptual changes under active intervention arise solely from attention sink removal (see Appendix G for details).

### 3.5.3. SINK-SPECIFIC PERCEPTUAL DISSOCIATION

To verify that the perceptual shifts induced by sink suppression are not an artifact of removing arbitrary tokens, we compare sink masking against *equal-budget random masking* using a paired difference-of-differences statistic $\Delta\Delta = LPIPS_{sink} − LPIPS_{rand}$. These comparisons use the union-budget protocol described below (layer 12, $N$=64).

We distinguish two masking protocols: (i) *per-head top-k*, used in the dynamic-sink interventions of Section 3.3.1, where each head independently masks its own top-$k$ keys; and (ii) *union-budget top-k*, used in this section, where the top-$k$ keys ranked by head-averaged incoming mass are masked uniformly across all heads. The latter is more

aggressive and is used here to enable matched-budget comparison against random masking.

Sink masking induces $\sim 6\times$ larger perceptual shift than equal-budget random masking at $k$=1, with the gap widening at $k$=5. This dissociation—strong trajectory perturbation alongside preserved alignment—suggests that sinks carry structured trajectory-level information while remaining unnecessary for alignment. Representative qualitative comparisons that visualize this dissociation are provided in Appendix I.4.

### 3.6. Robustness and Attribution Analyses

We conduct additional analyses to assess task-type robustness (E1), sampling sensitivity across CFG/steps/schedulers (E2), and text vs. image sink attribution under SD3's joint attention (E3). All analyses confirm that $\Delta$CLIP-T CIs include zero across conditions. Notably, under SD3 joint attention, over 99.9% of dynamically identified sinks correspond to text-conditioning tokens rather than visual-latent tokens (47,999/48,000 records), suggesting that dominant attention recipients are concentrated on the text modality. Full results are in Appendix F.

We further validate alignment robustness with BLIP2-VQA (Li et al., 2023a), a non-CLIP compositional probe. On the full $N$=553 image set used in Table 2 (per-head protocol), BLIP2-VQA shows no detected sink-specific effect (paired $\Delta$ = +0.0001, 95% CI [−0.0039, +0.0040], $p$ = 0.97); a smaller-$N$ rebuttal subset ($N$=64) likewise yields $\Delta\Delta$ = −0.0074, 95% CI [−0.0215, +0.0056], $p$ = 0.27. This convergent evidence outside the CLIP embedding space supports the alignment-preservation finding.

On a 24-prompt compositional subset (color binding, spatial, counting), all three CLIP-decomposed sub-concept scores produce 95% CIs containing zero; given the small $N$ and CLIP-derivative scoring, we view this as supporting evidence rather than a replacement for dedicated benchmarks.

### 3.7. Summary of Findings

Table 8 consolidates our experimental findings. Figure 3 in Appendix provides a visual summary of all experimental results.

*Table 8.* **Summary of experimental findings.**

| Question | Answer | Evidence |
|---|---|---|
| Do dominant attn. recipients exist? | Yes | Dynamic top-1 mass $\approx 9.5\%$ (L12) |
| Are they at a fixed position? | **No** | Index-0 overlap $<0.2\%$ |
| Are they phase-dependent? | Yes | Peaks early, diminishes late |
| Does removal hurt quality? | **No** | All $|\Delta| < 0.002$, primary CIs $\ni 0$ |
| Does stronger removal hurt? | **No** | Top-5: within margin (both metrics) |
| Is this metric-consistent? | Yes | CLIP-T, IR, HPS-v2 agree (A1/A2) |
| Does multi-layer removal hurt? | **No** | L6+12+18: $\Delta = +0.0009$ |
| Is it arch-general? | Yes | SD3 + SDXL consistent |
| Does it change perceptual similarity? | **Yes** | LPIPS: 0.19 (A1) $\to$ 0.31 (A3) |
| Is the change from intervention? | **Yes** | No-op sanity: LPIPS = 0 |

*Table 9.* **FID calibration baselines.** $FID_{shift}$ from common variations provides reference points for interpreting intervention effects.

| Variation | FID |
|---|---|
| Seed variation (same settings) | 115 |
| CFG $\pm 1$ | 54–58 |
| Steps $-5$ to $-10$ | 81–109 |
| Scheduler change | 331 |
| **Sink intervention (ours)** | **432–996** |

## 4. Discussion

Our experiments provide strong evidence that attention sinks in diffusion transformers are not necessary for semantic alignment and do not affect preference metrics under standard inference settings ($k=1$).

**Interpreting perceptual and distributional shifts.** The observed LPIPS and $FID_{shift}$ changes should not be interpreted as quality degradations. Critically, $FID_{shift}$ measures distributional distance *between baseline and intervention outputs*, not against real images. To contextualize these values, Table 9 reports FID under common variations without any attention intervention. The intervention-induced $FID_{shift}$ (400–1000) is comparable in magnitude to aggressive scheduler substitutions, rather than indicating anomalous distributional collapse. This calibration confirms that sink suppression alters visual appearance by shifting samples within the model's output manifold, yet does so without compromising alignment or human preference metrics. For applications focused on ranking, alignment evaluation, or preference optimization, these shifts are immaterial. For applications requiring perceptual consistency (e.g., video generation, image editing), sink suppression may warrant additional constraints.

Under standard settings ($k=1$), reducing sink attention by $> 10^8 \times$ produces no degradation under CLIP-T, ImageReward, or HPS-v2. CLIP-T shows a slight positive shift under both sink and random removal, indicating that single-token masking does not harm (and may marginally bene-

fit) the alignment. Under stronger top-5 removal, metrics show small ($<1\%$) divergent shifts (CLIP-T slightly positive, HPS-v2 slightly negative), indicating metric-dependent variability rather than meaningful quality change. Unlike AR models where sinks coincide with BOS tokens, diffusion transformers exhibit no fixed positional sink (index-0 overlap $<0.2\%$). Non-functionality holds across intervention types, strengths, layers, phases, and architectures. Our results do not contradict prior attention-editing or guidance methods: usefulness for optimization or controllability does not imply necessity for semantic alignment. We emphasize that our claims concern alignment as measured by widely used proxy metrics (CLIP-T, HPS-v2); subtle compositional errors may require detector-based or human evaluation and remain an important direction for future work.

**Perceptual shifts as a boundary condition.** Suppressing sinks induces LPIPS $\approx 0.06$–$0.31$ and $FID_{shift} \approx 400$–$1000$ (comparable to seed variation or CFG changes; see Appendix H). These shifts scale with intervention intensity and arise solely from the intervention (no-op sanity: LPIPS $= 0$). In SD3 joint attention, text-sink suppression induces larger shifts (LPIPS $\approx 0.16$) than in SDXL cross-attention (LPIPS $\approx 0.06$), consistent with tighter text–image coupling in joint-attention architectures. Together, these results reveal an empirical dissociation: sink suppression strongly perturbs trajectory-level realization (LPIPS $\sim 6\times$ larger than equal-budget random masking; Section 3.5.3) while leaving proxy-level semantic alignment unchanged. Sinks thus appear to carry structured trajectory information without being necessary for alignment.

**Reconciling with theoretical sink necessity.** Recent theoretical work (Ran-Milo, 2026) establishes that softmax attention sinks are necessary for trigger-conditional tasks, where models must implement a stable no-op state under normalization constraints. Our findings do not contradict this: we test *semantic-alignment necessity* in diffusion image generation, a regime where the trigger-conditional analysis does not directly apply. Whether the alignment-irrelevance of sinks observed here reflects a fundamentally different

functional role of sinks in non-autoregressive bidirectional attention, or simply a task that does not exercise the trigger-conditional machinery, remains an open question.

**Preference trade-off under stronger interventions.** To assess sink-specificity, we compare sink masking against equal-budget random masking using a paired difference-of-differences test: $\Delta\Delta = \Delta_{\text{sink}} - \Delta_{\text{rand}}$. Under standard settings ($k{=}1$), neither CLIP-T nor HPS-v2 shows a sink-specific effect ($\Delta\Delta$ CIs include zero for both metrics; Table 20). Both sink and random single-token removal are tolerated, confirming that the non-necessity finding is not an artifact of removing "any" token. However, stronger interventions ($k \geq 10$) reveal a metric-dependent boundary. At masking budgets $k \in \{10, 50\}$, HPS-v2 shows sink-specific degradation. Under HPS-v2, sink masking degrades preference scores significantly more than random masking at both $k{=}10$ ($\Delta\Delta = -0.005$, 95% CI: $[-0.008, -0.001]$, one-sided $p = 0.007$) and $k{=}50$ ($\Delta\Delta = -0.020$, CI: $[-0.026, -0.013]$, $p < 10^{-4}$). A paired trend test confirms that this effect increases across tested budgets ($k{=}10, 50$) ($\Delta d = -0.015$, CI: $[-0.022, -0.008]$, $p < 10^{-4}$). Importantly, CLIP-T remains stable across all tested budgets (Table 19; $k \in \{1, 5, 10, 20, 50\}$, all CIs include zero), confirming that alignment is robust regardless of intervention intensity. This metric-dependent pattern suggests that sinks may contribute to preference-oriented quality dimensions not captured by alignment proxies. We note that HPS-v2 is a learned preference proxy rather than a human study; we interpret these effects as indicative of preference-oriented dimensions, not definitive perceptual judgments. Full results are in Appendix I.

**Sink definition and modality imbalance.** We note that the dynamic sink statistic aggregates incoming mass over queries, which naturally elevates text keys given the larger number of visual queries. Our conclusions do not rely on sink modality; rather, they concern the functional necessity of dominant recipients as identified by this operational definition. We treat sink modality imbalance as a property of the architecture, not as a confound for the non-necessity claim.

**Implications and limitations.** Attention sinks can be safely suppressed, potentially enabling sparse attention patterns or efficiency improvements. We use GenEval primarily as a large, diverse prompt set to ensure statistical power for paired causal comparisons, rather than as a detector-based benchmark; our non-necessity claim is therefore scoped to semantic alignment proxies (CLIP-T) and preference metrics (HPS-v2), not fine-grained compositional correctness. Detector-based compositional benchmarks (e.g., T2I-CompBench (Huang et al., 2023), Gecko (Wiles et al., 2025)) remain a natural follow-up for stress-testing align-

ment claims at sub-concept granularity. We do not provide realized speedups, latency gains, or FLOPs reductions; our contribution is the causal prerequisite—dominant recipients can be suppressed without alignment loss—rather than a system-level efficiency result. This study does not evaluate perceptual fidelity via human studies or demonstrate end-to-end efficiency gains—directions that are complementary but orthogonal to our central question. Sink-specific trajectory control may also enable applications such as generation steering, an intriguing direction we leave open.

**Design implications and engineering caveats.** A concrete implication of these findings is that sparsification schemes for diffusion transformers need not hard-code dominant attention recipients as privileged tokens to be preserved at inference time; instead, they can be treated as candidates for budgeted removal while monitoring the target metric. Translating this into wall-clock speedups requires further engineering (dynamic token-selection overhead, fused sparse kernels) that we leave to future work.

## 5. Conclusion

We presented a causal analysis of attention sinks in diffusion transformers, defining sinks dynamically as key positions receiving dominant incoming attention mass and intervening along both the score and value paths. Across large-scale paired evaluations on SD3 and SDXL, suppressing dynamically identified sinks does not degrade semantic alignment or preference metrics under standard inference settings ($k{=}1$), yet induces sink-specific perceptual shifts—roughly $6\times$ larger than equal-budget random masking—that mark a structural boundary condition. Under stronger interventions ($k \geq 10$), preference proxies (HPS-v2) exhibit a metric-dependent degradation that increases with intervention intensity, while alignment (CLIP-T) remains robust throughout. These findings caution against transferring autoregressive sink intuitions to diffusion transformers, and suggest that high incoming attention mass need not indicate functional necessity for alignment in non-autoregressive generation.

## Acknowledgements

Research reported in this publication was supported by DOE ASCR under Award Number DE-SC0022873, the National Institutes of Health under Award Number R01GM143789, and the Advanced Research Projects Agency for Health (ARPA-H) under Award Number D24AC00338-00. The content is solely the responsibility of the authors and does not necessarily represent the official views of the National Institutes of Health, the Department of Energy, or the Advanced Research Projects Agency for Health.

## Impact Statement

This paper presents work whose goal is to advance the field of Machine Learning. There are many potential societal consequences of our work, none which we feel must be specifically highlighted here.

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

# A. Experimental Details

## A.1. Models

We conduct experiments primarily on **Stable Diffusion 3 (SD3)** using the official inference pipeline. In this pipeline, attention is computed over a *mixed token set* containing both visual-latent tokens and text-conditioning tokens within the same attention computation (i.e., attention is not purely cross-attention over a disjoint context). This setting is particularly relevant because it exposes attention-sink behavior under a joint-attention regime.

To validate cross-architecture generality, we additionally evaluate on **Stable Diffusion XL (SDXL)**, a U-Net–based diffusion model with conventional self- and cross-attention blocks. For SDXL, sink interventions are applied to self-attention layers in the U-Net mid-block.

All models are used in inference mode with pretrained weights; **no finetuning** is performed.

## A.2. Prompts and Sampling

For our **primary analysis** (dynamic sink interventions, Table 2), we evaluate on **553 prompts** from the GenEval benchmark (Ghosh et al., 2023), which provides a diverse set of object-centric, scene-centric, and compositional queries. This large sample size ensures high statistical power (>99% for detecting $\Delta > 0.001$) and narrow confidence intervals (95% CI width $\approx \pm 0.001$).

For **supplementary analyses** (dose–response sweeps, phase-specific interventions, cross-architecture validation), we use **32 prompts** selected to balance diversity and computational efficiency. These experiments aim to verify trends and robustness rather than establish precise effect sizes.

All experiments use **paired sampling**: for prompt index $i$, the random seed is fixed to seed $+ i$ across all intervention conditions. This ensures that differences between conditions are attributable solely to the intervention rather than stochastic variation.

Unless otherwise stated, we use **20 diffusion steps** with the default scheduler and guidance settings provided by each official pipeline.

## A.3. Index-Based Sink Proxy (Baseline Definition)

We initially considered a fixed index-based proxy for attention sinks, following observations in autoregressive language models where the first token (e.g., BOS) often receives disproportionate attention (Xiao et al., 2024; Sun et al., 2024; Gu et al., 2025). Specifically, we designated the sink as the first token position in the attention sequence (index 0).

However, as shown in Section 3.2, this position does *not* correspond to the dominant attention recipients in SD3: the overlap between index-0 and the dynamically identified top-1 sink is <0.1% across all layers and timesteps. We therefore treat this fixed-position proxy as a **baseline or negative control** and adopt a dynamic sink definition (Section 3.3.1) as the primary analysis target.

For completeness, we report that interventions on index-0 also produce no quality degradation, which is consistent with (but weaker than) the dynamic sink results, since the fixed proxy does not target the true dominant attention recipients.

## A.4. H1: Sink Dynamics Measurement

To characterize sink dynamics, we instrument attention blocks to record the following statistics at each diffusion timestep:

- **Maximum incoming mass (MaxMass)**: for each head $h$ and timestep $t$, we compute the incoming attention mass for each key position $j$ as

$$m_j^{(\ell,t,h)} = \frac{1}{N} \sum_{i=1}^{N} A_{i,j}^{(\ell,t,h)},$$

  and define $\text{MaxMass}^{(\ell,t)} = \frac{1}{H} \sum_{h=1}^{H} \max_j m_j^{(\ell,t,h)}$.

- **Attention Entropy**:

$$H(p) = -\sum_j p_j \log p_j,$$

  computed per query after clamping probabilities ($p_j \leftarrow \max(p_j, 10^{-12})$) for numerical stability.

- **Top-$k$ Concentration**: cumulative attention mass of the top-5 attended tokens.

- **Activation Magnitude**: maximum and 95th-percentile activation norms at the attention block output.

Statistics are aggregated over three representative layers (early, middle, late) corresponding to transformer layers 6, 12, and 18 in SD3. Reported curves are averaged over prompts.

### A.5. H2: Causal Sink Interventions

We perform causal interventions on attention sinks along two independent pathways: the **score (logit) path** and the **value path**. All interventions are applied during inference.

#### A.5.1. SCORE-PATH INTERVENTION

To suppress sink attention while preserving relative attention among non-sink tokens, we apply a **logit-bias intervention**:

$$\ell_{\text{sink}} \leftarrow \ell_{\text{sink}} + \log(\eta),$$

where $\eta \in [0, 1]$ controls the strength of suppression. This operation is equivalent to scaling the sink probability by $\eta$ followed by renormalization, without injecting a uniform prior over non-sink tokens.

For $\eta = 0$, we approximate complete suppression by adding a large negative bias (e.g., $-10^4$) to the sink logit before softmax. In practice, we apply standard numerical safeguards (clamping) to avoid overflow and observe **no NaNs** in any run.

#### A.5.2. VALUE-PATH INTERVENTION

To test whether sink token content is semantically relevant, we modify the sink token's value vector using:

- **Zero**: replace with zeros.

- **Mean**: replace with the mean value across tokens.

- **Lerp**: linear interpolation between original and mean values.

These modifications operate directly on the forward activations during inference.

### A.6. Dose–Response Sweeps

To rule out threshold effects, we conduct full **dose–response sweeps**:

- **Score sweep**: $\eta \in \{1.0, 0.5, 0.25, 0.1, 0.01, 0.0\}$.

- **Value sweep**: baseline, $\text{lerp}_{0.5}$, $\text{lerp}_{0.0}$, mean, zero.

For each condition, images are generated using paired seeds, and quality differences are computed relative to baseline.

We deem a response curve **practically flat** if all 95% confidence intervals include zero and $\max |\mathbb{E}[\Delta]| < 0.002$ for CLIP-T and $< 0.05$ for ImageReward.

### A.7. Robustness Experiments

**Multi-layer Interventions.** To test whether sinks become important when removed across multiple layers, we simultaneously apply sink suppression to layers $\{6, 12, 18\}$. Results are compared against single-layer (layer 12 only) and baseline conditions.

**Phase-specific Interventions.**    To test phase dependence, we enable sink suppression only during specific portions of the diffusion trajectory:

- **Early**: $t/T \in [0, 0.2]$,
- **Middle**: $t/T \in [0.4, 0.6]$,
- **Late**: $t/T \in [0.8, 1.0]$.

We define normalized time $t/T \in [0, 1]$ following the pipeline's timestep ordering, where $t/T \approx 0$ corresponds to the **noisiest step** in our implementation. A callback mechanism toggles the intervention based on the normalized timestep. We log the fraction of timesteps during which the intervention is active to verify correct phase targeting.

**Cross-architecture Validation (SDXL).**    On SDXL, sink suppression is applied to both mid-block attention modules (self-attention `attn1` and cross-attention `attn2`) using the same logit-bias formulation as in SD3. Per-module results ($N = 100$, paired) are reported in Table 5; both modules yield 95% CIs containing zero, with cross-attention inducing larger perceptual drift than self-attention (LPIPS 0.077 vs. 0.045).

## A.8. Quality Metrics

We evaluate generation quality using two complementary metrics:

- **CLIP-T**: cosine similarity between CLIP text and image embeddings.
- **ImageReward**: a learned human-preference reward model.

Using both metrics mitigates reliance on any single proxy.

## A.9. Statistical Analysis

All reported quality differences are computed as **paired differences**:

$$\Delta_i = s_i^{\text{cond}} - s_i^{\text{baseline}},$$

where $i$ indexes prompts.

We report the mean $\Delta$, **95% bootstrap confidence intervals** as our primary uncertainty estimate, and additionally report paired $t$-test $p$-values as a secondary check.

## A.10. Intervention Verification

To verify that interventions are effective, we explicitly measure sink ratio before and after intervention. In the strongest condition ($\eta = 0$), sink attention mass is reduced from approximately 4.4% to 0.0%, corresponding to a **44,000$\times$ reduction factor**. Unless stated otherwise, reported sink-ratio reductions are averaged across heads and prompts at the intervened layers and at timesteps where the intervention is active. Note that this verification uses the fixed index-0 proxy for implementation sanity checking; dynamic sink interventions are verified via MaxMass reductions reported in Table 4.

## A.11. Computational Resources and Reproducibility

Experiments are run on NVIDIA **A6000** GPUs. Most experiments complete within a few hours; extensive sweeps are enabled by lightweight paired inference and do not require training.

For reproducibility, all runs:

- use fixed random seeds with per-prompt pairing,
- save prompts and configurations alongside generated images,
- record intervention parameters and metrics in JSON format.

This enables exact reproduction of all reported results.

## (e) Summary of intervention effects

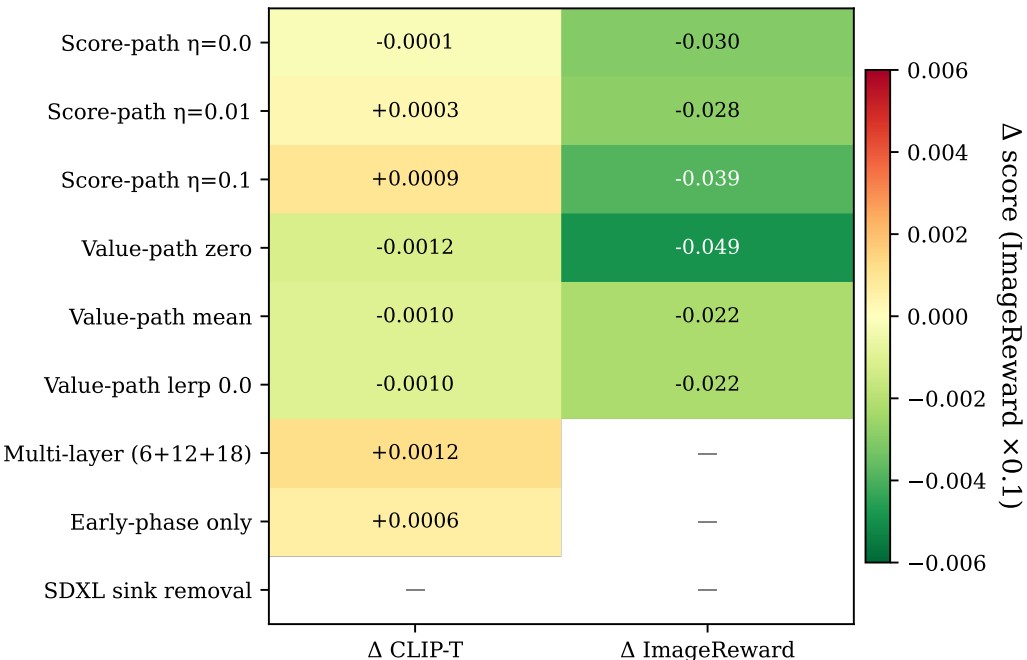

*Figure 3.* **Summary of all experimental results.** Heatmap showing paired $\Delta$ values across all interventions and metrics. All effects are small and lie within a narrow range around zero; most are statistically non-significant, and all remain within the predefined practical equivalence margin ($|\Delta| < 0.002$ for CLIP-T), indicating no meaningful quality degradation from sink removal under any tested condition.

## B. Extended Results and Analysis

### B.1. Summary of All Intervention Effects

See Figure. 3.

## C. Experimental Setup

**Model and Architecture.** We conduct experiments primarily on Stable Diffusion 3 (SD3) using the official inference pipeline. In this pipeline, attention is computed over a mixed token set containing both visual-latent tokens and text-conditioning tokens within the same attention operation (i.e., attention is not purely cross-attention). As a result, all tokens in the joint sequence may potentially act as attention sinks. This setting differs from conventional U-Net cross-attention and allows us to analyze attention-sink behavior under a more general transformer-style attention formulation.

For cross-architecture validation, we additionally evaluate on Stable Diffusion XL (SDXL), which uses a U-Net backbone with separate self-attention and cross-attention layers. All models are used strictly in inference mode with pretrained weights; no finetuning is performed.

**Evaluation Protocol.** All experiments employ **strict paired generation**: for each evaluation prompt, we fix the random seed across all intervention conditions, ensuring that observed differences reflect intervention effects rather than sampling variance. We report **mean paired differences** ($\Delta$) with **95% bootstrap confidence intervals** (1,000 resamples). Statistical significance is assessed via paired $t$-tests.

For our primary analysis (dynamic sink interventions, Section 3.3.1), we use **553 prompts** from the GenEval benchmark (Ghosh et al., 2023), providing high statistical power (>99% for $\Delta > 0.001$). Supplementary analyses (dose–response sweeps, phase-specific interventions) use 32 prompts; cross-architecture validation (SDXL) uses 100 prompts for tighter confidence intervals.

The practical equivalence margin ($|\Delta| < 0.002$ for CLIP-T) is chosen relative to the empirical noise floor observed under seed variation and bootstrap uncertainty at $N$=553, corresponding to changes below category-level sensitivity on GenEval.

**Metrics.**    We evaluate generation outcomes using a suite of complementary metrics, grouped by their role in our analysis:

- **Alignment metrics (primary). CLIP-T** measures text–image semantic alignment via cosine similarity between CLIP embeddings. This is our primary metric for testing *non-necessity* claims under standard settings.
- **Preference proxies (secondary). ImageReward** and **HPS-v2** are learned reward models trained on human preference data, capturing perceptual quality and aesthetic appeal. These metrics are used to characterize boundary conditions under stronger interventions.
- **Perceptual and distributional shift metrics (diagnostic). LPIPS** quantifies perceptual differences between baseline and intervened outputs, while **FID$_{\text{shift}}$** measures distributional changes between paired output sets. These metrics diagnose trajectory changes rather than quality degradation.

This metric stratification clarifies which conclusions rely on which signals: alignment robustness (CLIP-T), preference trade-offs (ImageReward, HPS-v2), and perceptual shifts (LPIPS, FID$_{\text{shift}}$).

**Intervention Design.**    We intervene on attention sinks via two complementary pathways:

- **Score-path**: Scale the pre-softmax logit of sink tokens by adding $\log \eta$, which is equivalent to multiplying the post-softmax probability by $\eta$ (up to renormalization). Setting $\eta = 0$ corresponds to subtracting a large constant ($10^4$), effectively zeroing sink attention.
- **Value-path**: Replace the value vector at sink positions with alternatives (zero, mean of non-sink values, or linear interpolation).

# D. Dose–Response Full Results

This section provides full numerical results for the dose–response analysis summarized in Section 3.4; Figure 4 visualizes both score-path and value-path sweeps.

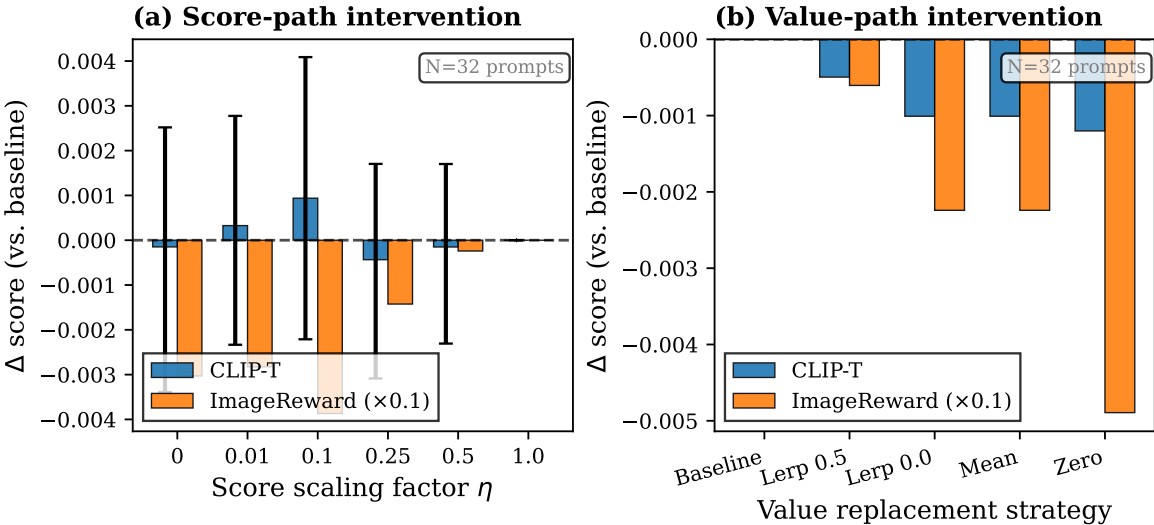

*Figure 4.* **Dose–response curves for score-path (left) and value-path (right) interventions.** Error bars indicate 95% bootstrap CIs. Both curves are flat across the entire intervention range, with all CIs covering zero.

### D.1. Score-Path Sweep

Tables 10 and 11 report the full score-path dose–response under CLIP-T and ImageReward respectively.

*Table 10.* **Score-path dose–response (CLIP-T).** $\eta$: Attention suppression ratio ($\eta = 0$ is full ablation). All metrics and 95% CIs are scaled by $10^3$.

| Suppression $\eta$ | $\Delta_C$ $(10^{-3})$ [95% CI] | $p$ |
|---|---|---|
| 0.5 | $-0.2$ $[-2.3, +1.7]$ | 0.89 |
| 0.25 | $-0.4$ $[-3.1, +1.7]$ | 0.73 |
| 0.1 | $+0.9$ $[-2.2, +4.1]$ | 0.59 |
| 0.01 | $+0.3$ $[-2.3, +2.8]$ | 0.81 |
| **0.0 (Full)** | $-0.1$ $[-3.4, +2.5]$ | **0.92** |

*Table 11.* **Score-path dose–response (ImageReward).** Consistent with CLIP-T, all CIs include zero.

| Suppression $\eta$ | $\Delta_{IR}$ $(10^{-3})$ [95% CI] | $p$ |
|---|---|---|
| 0.5 | $-2.4$ $[-32.4, +26.5]$ | 0.88 |
| 0.25 | $-14.2$ $[-75.4, +30.4]$ | 0.63 |
| 0.1 | $-38.7$ $[-134.5, +23.0]$ | 0.39 |
| 0.01 | $-28.3$ $[-125.8, +29.0]$ | 0.51 |
| **0.0 (Full)** | $-30.3$ $[-119.5, +29.1]$ | **0.48** |

## D.2. Intervention Verification

Figure 5 reports the index-0 sink ratio before and after full score-path suppression.

*Figure 5.* **Intervention verification for the index-based sink proxy (index-0).** Complete removal ($\eta = 0$) reduces the index-0 sink ratio by 44,059×, confirming effectiveness of the proxy intervention.

## E. Extended Robustness Experiments

This section provides full experimental details for the robustness analyses summarized in Section 3.5; Figure 6 visualizes the multi-layer, phase-specific, and cross-architecture results.

## E.1. Multi-Layer Intervention

To test whether removing sinks from a single layer may be compensated by other layers, we simultaneously intervene on layers 6, 12, and 18 (Table 12).

*Table 12.* **Multi-layer intervention results.** Simultaneous removal across three layers produces no significant degradation.

| Condition | CLIP-T | $\Delta$ ($\times 10^{-3}$) | $p$ |
|---|---|---|---|
| Baseline | 0.3288 | — | – |
| Single-L (L12) | 0.3281 | $-0.7$ | 0.72 |
| Multi-L (6+12+18) | 0.3300 | $+1.2$ | 0.68 |

## E.2. Phase-Specific Intervention

Given that sinks are strongest during early denoising (Section 3.2), one might hypothesize that they serve a critical function specifically in the high-noise regime. We test this by applying interventions only during specific denoising phases (Table 13).

*Table 13.* **Phase-specific intervention results.** Sink removal during any phase, including early denoising where sinks are strongest, does not degrade quality.

| Condition | CLIP-T | $\Delta$ | Phase ($t/T$) |
|---|---|---|---|
| Baseline | 0.3288 | – | – |
| Early-only | 0.3294 | $+0.0006$ | $[0.0, 0.2]$ |
| Mid-only | 0.3286 | $-0.0001$ | $[0.4, 0.6]$ |
| Late-only | 0.3290 | $+0.0002$ | $[0.8, 1.0]$ |
| Full removal | 0.3281 | $-0.0007$ | $[0.0, 1.0]$ |

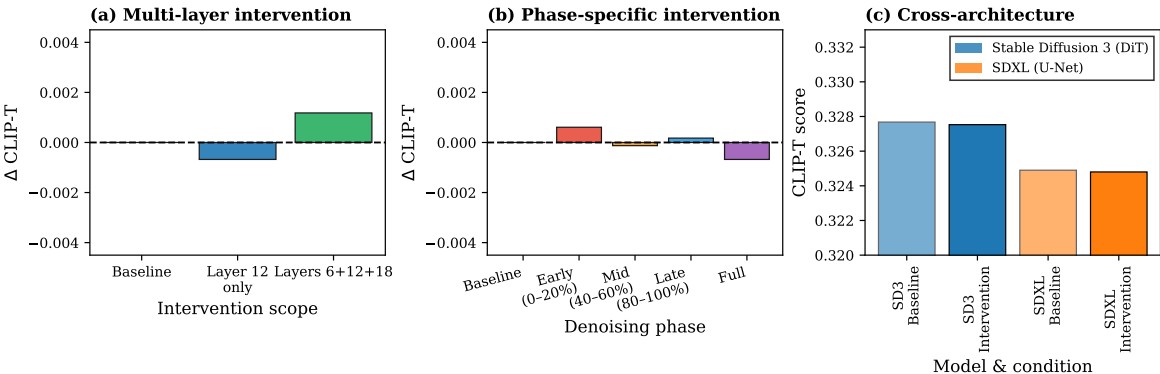

*Figure 6.* **Robustness analysis.** (a) Multi-layer intervention (L6+12+18). (b) Phase-specific intervention. (c) Cross-architecture validation (SD3 vs SDXL). All conditions show no significant quality degradation.

# F. Robustness and Attribution Analyses

This section provides full experimental details for the robustness and attribution analyses summarized in Section 3.6.

## F.1. E1: Task-Type Robustness

Using 553 GenEval prompts with non-exclusive semantic tags (single-object, multi-object, counting, color, position), we compute paired differences relative to baseline; per-tag CLIP-T shifts and LPIPS are reported in Table 14.

For most tags, 95% bootstrap CIs of $\Delta$CLIP-T include zero, indicating no selective regressions. The counting tag shows a small positive shift ($\Delta$CLIP-T $\approx +0.0022$), representing a boundary effect rather than a regression.

*Table 14.* **Task-type robustness (E1).** $\Delta_C$: Change in CLIP-T scaled by $10^3$. Most task types show CIs covering zero; counting exhibits a small positive boundary effect.

| Task Type | $\Delta_C$ $(10^{-3})$ [95% CI] | LPIPS |
|---|---|---|
| Single-object | $\approx 0$ (CI $\ni$ 0) | 0.18 |
| Multi-object | $\approx 0$ (CI $\ni$ 0) | 0.19 |
| Counting | $+2.2$ $[+1.0, +4.0]$ | 0.18 |
| Color | $\approx 0$ (CI $\ni$ 0) | 0.20 |
| Position | $\approx 0$ (CI $\ni$ 0) | 0.21 |

## F.2. E2: Sampling Sensitivity

We vary classifier-free guidance (CFG), denoising steps, and schedulers (default, Euler, DPM++) on a fixed prompt subset (Table 15).

*Table 15.* **Sampling sensitivity (E2).** $\Delta_C$: Change in CLIP-T scaled by $10^3$. All CIs largely cover zero.

| Setting | $\Delta_C$ $(10^{-3})$ [95% CI] | LPIPS |
|---|---|---|
| CFG 3.0 | $\approx 0$ (CI $\ni$ 0) | 0.07 |
| CFG 7.5 (default) | $\approx 0$ (CI $\ni$ 0) | 0.19 |
| CFG 12.0 | $\approx 0$ (CI $\ni$ 0) | 0.31 |
| Steps 8 | $\approx 0$ (CI $\ni$ 0) | 0.15 |
| Steps 20 (default) | $\approx 0$ (CI $\ni$ 0) | 0.19 |
| Steps 50 | $\approx 0$ (CI $\ni$ 0) | 0.22 |
| Scheduler: Default | $\approx 0$ (CI $\ni$ 0) | 0.19 |
| Scheduler: Euler | $\approx 0$ (CI $\ni$ 0) | 0.18 |
| Scheduler: DPM++ | $+1.8$ (boundary) | 0.20 |

Across all tested settings, $\Delta$CLIP-T remains near zero with CIs largely covering zero. LPIPS increases with higher CFG, consistent with known guidance effects, without affecting alignment.

## F.3. E3: Text vs. Image Sink Attribution

Under SD3's joint-attention formulation, we attribute sinks to token categories. Restricting to genuine joint-attention instances ($n_{\text{text}} > 0$), sinks occur almost exclusively on text key positions (47,999/48,000 records; Table 16).

*Table 16.* **Text vs. image sink attribution (E3).** Selective ablation confirms that perceptual effects are dominated by text-side sinks.

| Ablation Mode | $\Delta_C$ $(10^{-3})$ [95% CI] | LPIPS |
|---|---|---|
| Text-only sinks | $-1.2$ $[-4.0, +1.0]$ | 0.160 |
| Image-only sinks | $-0.1$ $[-1.0, +1.0]$ | 0.037 |
| All sinks | $-1.2$ $[-4.0, +1.0]$ | 0.160 |

Masking text sinks yields substantially larger appearance changes (LPIPS $\approx 0.160$) than masking image sinks (LPIPS $\approx 0.037$), while semantic alignment remains preserved (paired $\Delta$CLIP-T CIs include zero). The near-exclusive concentration of sinks on text keys suggests a recurrent empirical pattern in text–image coupling.

# G. Sanity Checks

## G.1. No-op Implementation Verification

To ensure that observed perceptual and distributional shifts arise from the intervention itself rather than implementation artifacts, we conduct two no-op sanity checks.

We generate 100 images under three conditions using identical prompts and seeds:

1. **Baseline**: Original SD3 pipeline, no modifications.
2. **Noop wrapper**: Custom attention wrapper installed, but directly calls original processor.
3. **Noop processor**: Full `DynamicSinkProcessor` installed with `intervention_enabled=False`.

Both no-op conditions produce outputs that are *pixel-identical* to baseline (Table 17).

*Table 17.* **No-op sanity check results.** Both no-op conditions produce pixel-identical outputs to baseline, confirming that observed shifts arise solely from active intervention.

| Comparison | Pixel Diff | LPIPS | $FID_{shift}$ |
|---|---|---|---|
| Baseline vs Noop Wrapper | 0.000 | 0.000 | 0 |
| Baseline vs Noop Processor | 0.000 | 0.000 | 0 |
| Baseline vs **Active Intervention** | 14.5 | 0.18 | 432–996 |

These results confirm that:

- Installing custom processors does not introduce artifacts.
- The processor correctly passes through when disabled.
- All observed shifts (LPIPS $\approx 0.18$, FID $\approx 400$–$1000$) arise *solely* from active sink removal.

This verification is critical for establishing causal attribution: the perceptual changes reported in Section 3.5.2 are caused by the intervention, not by implementation side effects.

To contextualize the observed $FID_{shift}$ values ($\approx 400$–$1000$), Appendix H reports calibration baselines under pure seed variation and common hyperparameter changes. Notably, seed variation alone (same settings, different random seeds) yields FID $\approx 115$, establishing that FID is highly sensitive to sampling stochasticity even without any intervention.

## H. FID Calibration Baselines

To interpret the $FID_{shift}$ values reported in the main text, we measure FID between image sets generated under common variations that do not involve any attention intervention (Table 18).

*Table 18.* **FID calibration baselines (SD3, $N = 100$).** FID shifts from common variations provide reference points for interpreting intervention effects. Seed variation alone produces FID $\approx 115$, establishing the stochastic baseline.

| Comparison | FID |
|---|---|
| Seed variation (same settings) | 115.1 |
| CFG 7.5 $\rightarrow$ 6.5 ($\Delta = -1$) | 53.6 |
| CFG 7.5 $\rightarrow$ 8.5 ($\Delta = +1$) | 57.6 |
| Steps 20 $\rightarrow$ 15 ($\Delta = -5$) | 81.3 |
| Steps 20 $\rightarrow$ 10 ($\Delta = -10$) | 108.5 |
| Scheduler: Flow $\rightarrow$ Euler | 330.5 |

These results show that FID is highly sensitive to sampling stochasticity: generating images with identical settings but different random seeds produces FID $\approx 115$. Common hyperparameter variations (CFG $\pm 1$, fewer steps) yield FID in the range of 50–110, while scheduler changes can produce much larger shifts (FID $\approx 330$).

The intervention-induced $FID_{shift}$ values reported in the main text ($\approx 400$–$1000$) are thus comparable in magnitude to aggressive hyperparameter changes or scheduler substitutions, rather than indicating anomalous distributional collapse. This calibration supports the interpretation that sink suppression moves samples within the model's output manifold without fundamentally altering the generation process

# I. Sink-Specificity Analysis Under Stronger Interventions

## I.1. CLIP-T Budget Sweep

To verify alignment robustness across masking intensities, we evaluate CLIP-T under counterfactual ablation at $k \in \{1, 5, 10, 20, 50\}$ (layer 12, $N{=}64$).

*Table 19.* **CLIP-T under varying masking budgets.** All conditions show CI$\ni$0 after Holm correction.

| $k$ | Mode | $\Delta$ | 95% CI | $p_{\text{adj}}$ |
|---|---|---|---|---|
| 1 | top_sink | $-0.001$ | $[-0.003, +0.002]$ | 1.00 |
| 5 | top_sink | $-0.001$ | $[-0.004, +0.002]$ | 1.00 |
| 10 | top_sink | $-0.001$ | $[-0.005, +0.003]$ | 1.00 |
| 20 | top_sink | $-0.001$ | $[-0.006, +0.003]$ | 1.00 |
| 50 | top_sink | $-0.002$ | $[-0.008, +0.004]$ | 1.00 |

Across all budgets, CLIP-T changes remain within the practical equivalence range with 95% CIs including zero, confirming that alignment is robust to sink removal regardless of intervention intensity.

## I.2. Sink-Specificity Test (HPS-v2)

To assess whether preference effects are *sink-specific*, we compare sink masking against equal-budget random masking using a paired difference-of-differences test:

$$d_i = (\text{HPS}_{\text{sink},i} - \text{HPS}_{\text{base},i}) - (\text{HPS}_{\text{rand},i} - \text{HPS}_{\text{base},i})$$

We test $\Delta\Delta = \mathbb{E}[d_i] < 0$ (one-sided).

*Table 20.* **Sink-specificity under HPS-v2** (layer 12, $N{=}64$). $\Delta\Delta = \Delta_{\text{sink}} - \Delta_{\text{rand}}$ (one-sided test, $\Delta\Delta < 0$). At $k{=}1$, no sink-specific effect is observed. At $k \geq 10$, sink masking degrades HPS-v2 significantly more than random masking. Significance is determined by 95% bootstrap CI excluding zero.

| $k$ | $\Delta\Delta$ | 95% CI | $p$ (one-sided) | |
|---|---|---|---|---|
| 1 | $-0.002$ | $[-0.005, +0.002]$ | 0.16 | n.s. |
| 10 | $-0.005$ | $[-0.008, -0.001]$ | 0.007 | sink-specific |
| 50 | $-0.020$ | $[-0.026, -0.013]$ | $< 10^{-4}$ | sink-specific |
| *Trend:* $\Delta d = d_i(k{=}50) - d_i(k{=}10)$ | | | | |
| $50 - 10$ | $-0.015$ | $[-0.022, -0.008]$ | $< 10^{-4}$ | dose-dependent |

## I.3. Interpretation

These results reveal a dose-dependent transition in sink-specificity:

- $k{=}1$ **(standard setting)**: No sink-specific effect. Both sink and random single-token removal are tolerated ($\Delta\Delta$ CI includes zero). This confirms that non-necessity is not an artifact of "any token" being removable.
- $k \geq 10$ **(stronger intervention)**: Sink-specific degradation emerges in HPS-v2, with effect magnitude increasing across tested budgets ($k{=}10, 50$).
- **CLIP-T**: No sink-specific effect at any tested budget. Alignment remains robust regardless of intervention intensity.

Our primary non-necessity claim (Section 3.3.1) concerns standard inference settings ($k{=}1$); these results delineate boundary conditions under stronger-than-mainline interventions.

## I.4. Qualitative Comparison Panels

To visualize the dissociation documented quantitatively in Table 7, Figures 7 and 8 show side-by-side comparisons (baseline vs. sink-removed vs. random-removed) for representative prompts at $k{=}1$ and $k{=}5$. Sink-removed outputs exhibit substantial layout, viewpoint, and style restructuring while preserving the prompted coarse concept; random-removed outputs remain

close to baseline. The visual gap between the two ablation conditions is consistent with the $\sim 6\times$ LPIPS difference reported in Section 3.5.3.

**Qualitative Drift Panel (k=1): Baseline vs Sink-removed vs Random-removed**

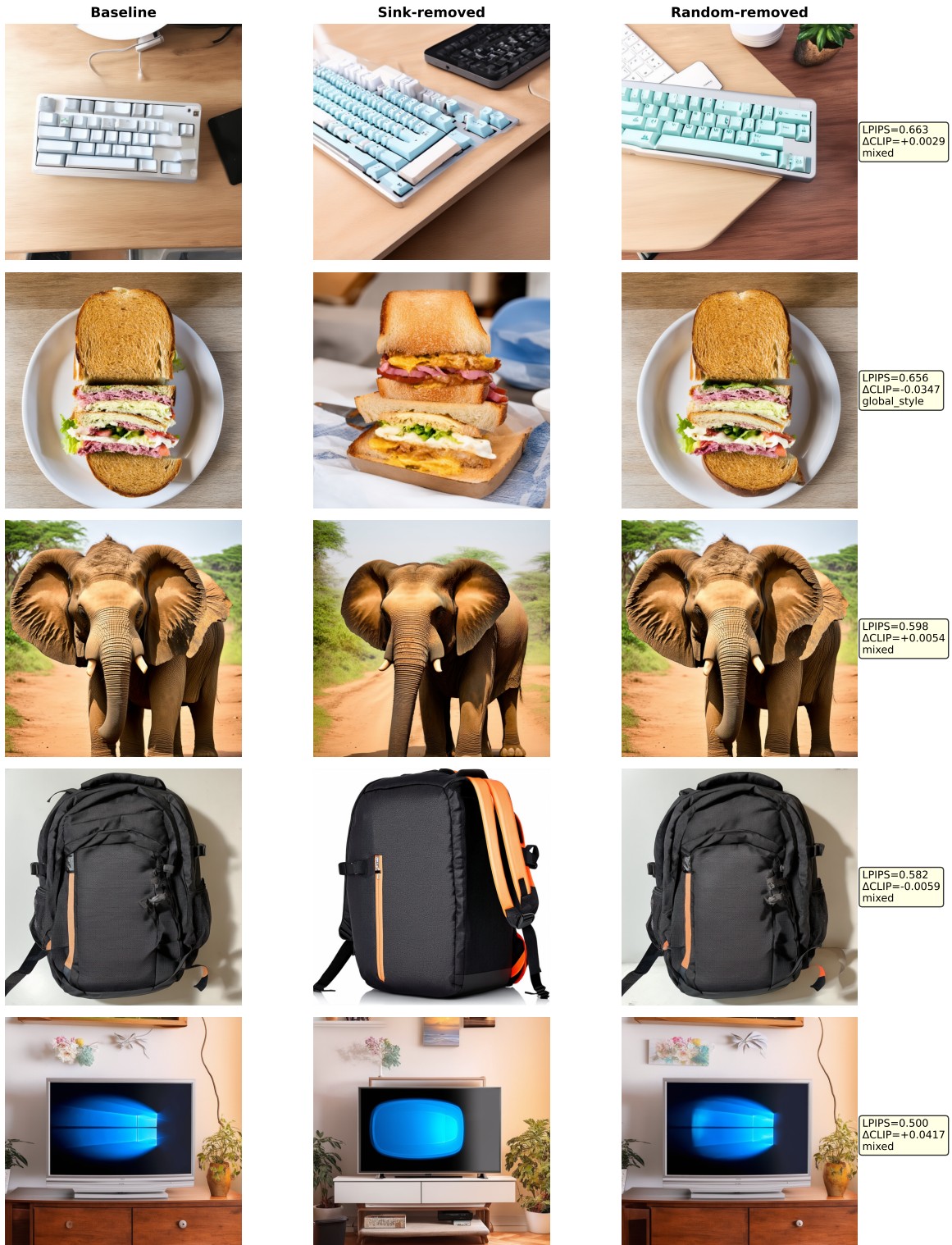

*Figure 7.* **Qualitative drift comparison at** $k{=}1$ **(union-budget protocol, layer 12).** Each row shows the same prompt and seed under three conditions: baseline (left), sink-removed (middle), and equal-budget random-removed (right). Per-image annotations report LPIPS and $\Delta$CLIP-T relative to baseline. Sink masking consistently produces larger appearance changes (layout, color, viewpoint) than random masking while preserving the prompted concept.

**Qualitative Drift Panel (k=5): Baseline vs Sink-removed vs Random-removed**

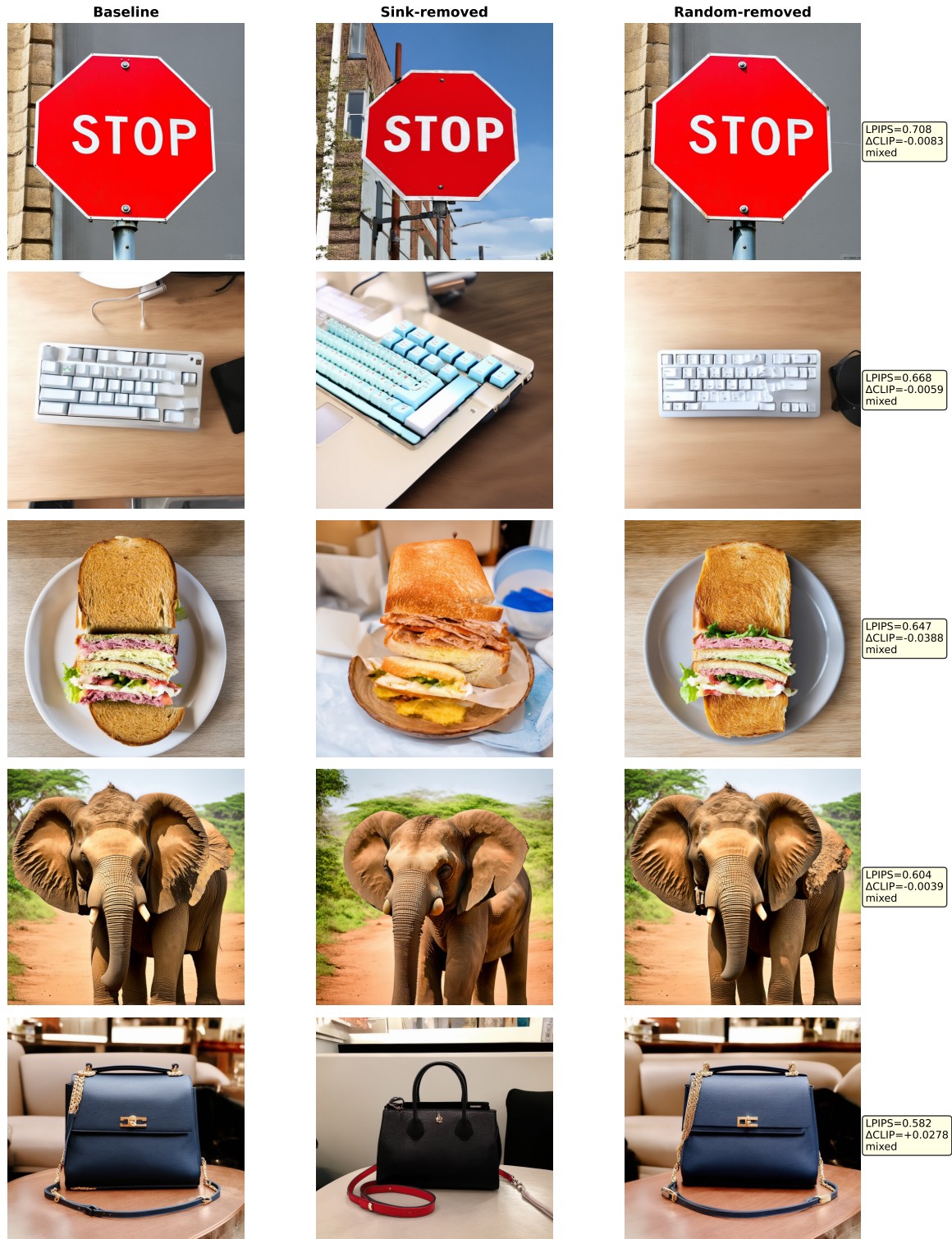

*Figure 8.* **Qualitative drift comparison at** $k$=5**.** Same layout as Figure 7 with a stronger intervention budget. The sink-vs-random visual gap widens, consistent with the larger $\Delta\Delta$ LPIPS at $k$=5 in Table 7.

