# OpenReview forum: "Attention Sinks in Diffusion Transformers: A Causal Analysis"
_ICML.cc/2026/Conference — ICML 2026 regular_

### Official Review · Reviewer_G5q6 · 2026-03-01

**Soundness:** 3
**Presentation:** 3
**Significance:** 2
**Originality:** 2
**Overall Recommendation:** 4
**Confidence:** 4

**Summary:**

This paper conducts a causal analysis of attention sinks in Diffusion Transformers (DiTs). While attention sinks have been found to be functionally important in autoregressive models, this paper poses the question if attention sinks are necessary for good generation results for DiTs. The authors define sinks dynamically as top-k key positions by incoming attention mass. Through causal interventions such as biasing down the attention logits/scores of sink key positions in the attention heads or replacing sink value projection features with substitutes, the paper finds that removing sinks does not hurt semantic alignment, although it does cause perceptual and distributional shifts. This suggests that sinks influence the sampling trajectory more than prompt adherence.

**Compliance With Llm Reviewing Policy:**

Affirmed.

**Final Justification:**

The main point in my rebuttal was the lack of the random masking baseline, which the authors addressed. My other main concern was the lack of an application, but I think is reasonable to leave for future work. Overall, I think the paper is sound and interesting.

**Key Questions For Authors:**

* Do the authors have any mechanistic insight into why sinks emerge in DiTs or what computation they support when present?
* How much do attention sinks influence sampling trajectories under LPIPS/FID_Shift relative to interventions on random tokens?
* Given that sinks are not necessary for alignment, how can this be applied to pruning, sparsifying attention, quantization, etc.? A small demo would make the paper more compelling.
* How do the findings contrast or support mechanistic explanations of attention sinks done on general Vision Transformers [2]?

[2] Jiang, N. et al. "Vision Transformers Don't Need Trained Registers." NeurIPS 2025.

**Limitations:**

Yes.

**Strengths And Weaknesses:**

**Strengths**
* The paper conducts a thorough set of causal interventions to measure the role of attention sinks in DiTs
* The paper is clearly written. Table 7, which summarizes the experimental findings, is very useful for the reader, and can even be introduced earlier.
* The finding that intervening on attention sinks does not degrade semantic alignment is surprising as Darcet et al. [1] and Jiang et al. [2] found that in ViTs, register/outlier tokens serve as a form of global memory.
* The results are validated on multiple models, generalizing the findings beyond Stable Diffusion 3

**Weaknesses**
* The main weakness of the paper is that it mainly falsifies prior hypotheses for the role of attention sinks in the DiT case but provides limited mechanistic insight into why sinks emerge in DiTs or what computation they support when present, beyond describing where and when they occur across layers and denoising phases.
* The finding that the sinks don't occur at the first token and are dynamic is limited in novelty as [1] already shows how the registers/sinks occur in different locations in ViTs
* The statement "attention sinks influence sampling trajectories" (L400-401) would be strengthened by quantifying how large this effect is relative to appropriate baselines. Interventions on random tokens and measuring the LPIPS/FIDshift under equal-budget random token masking would help establish sink-specificity for perceptual shifts. The paper already performs sink-specificity tests against equal-budget random masking for alignment/preference proxies, but it’s less clear whether the perceptual shifts are sink-specific or due to just masking  *some* tokens.
* Given the finding that sinks are not necessary for alignment, utilizing this for some application like pruning, sparsifying attention, quantization, etc. (as mentioned in the discussion) would make the paper's quality and impact higher.
* Missing reference to [2], which is relevant to mechanistically understanding attention sinks in vision models.

[1] Darcet, T. el al. "Vision Transformers Need Registers." ICLR 2024.

[2] Jiang, N. et al. "Vision Transformers Don't Need Trained Registers." NeurIPS 2025.

---

> ### Author Rebuttal · Authors · 2026-03-26
>
> We thank the reviewer for the specific questions, particularly the suggestion of a random-masking baseline. We have conducted this experiment and report the results below.
>
> **Perceptual sink-specificity (Q2, W3).** Equal-budget comparison (N=64, layer 12):
>
> | k | LPIPS (sink) | LPIPS (random) | $\Delta\Delta$ | 95% CI | p |
> |---|---|---|---|---|---|
> | 1 | 0.347 | 0.053 | **+0.295** | [+0.265, +0.323] | <0.0001 |
> | 5 | 0.436 | 0.104 | **+0.332** | [+0.308, +0.358] | <0.0001 |
>
> Sink masking induces $\sim$6.6$\times$ larger perceptual shift than random at equal budget. CLIP-T shows no detected sink-specific effect ($\Delta\Delta$ CI $\ni 0$). This dissociation suggests that sinks carry structured trajectory-level information while remaining unnecessary for alignment under the tested proxy metrics. More broadly, sink suppression strongly perturbs perceptual/trajectory realization while leaving proxy-level semantic alignment largely unchanged, suggesting an empirical dissociation between these two aspects in DiT attention. Representative triplet panels are provided at https://anonymous.4open.science/r/attention-sink-VIS-4C5B/rebuttal_drift_panel_k1.pdf (k=1) and https://anonymous.4open.science/r/attention-sink-VIS-4C5B/rebuttal_drift_panel_k5.pdf (k=5).
>
> **Mechanistic insight (W1, Q1).** Our paper primarily provides causal falsification. Empirical patterns point toward a partial mechanism: (1) sink concentration peaks during early denoising (Figure 2); (2) $>$99.9% of sinks are text tokens (E3); (3) text-side suppression produces larger perceptual shifts (LPIPS 0.160 vs 0.037). These are consistent with the hypothesis that sinks may act as transient text-conditioning anchors, but a full mechanistic account remains open.
>
> **Darcet et al. / Jiang et al. (W2, Q4).** Darcet (2024) and Jiang (2025) study registers in discriminative ViTs. We ask a different question: are dominant recipients *causally necessary* for generation in *diffusion transformers*? Our novelty is therefore not merely that sinks are dynamic, but that we causally test and falsify their necessity for proxy-level alignment in a generative diffusion setting. The finding that they are not under our tested setting is a non-obvious contrast to the functional importance of registers in discriminative ViTs. The revision will add a paragraph in §2 citing Jiang et al. and discussing this complementarity between discriminative register necessity and generative sink dispensability.
>
> **Pruning demo (W4, Q3).** We agree that a pruning or sparse-attention demo would strengthen the practical impact. Our current results establish the causal prerequisite: dominant recipients can be suppressed without detected alignment loss. Translating this into wall-clock speedups requires additional engineering (dynamic token selection overhead, fused sparse kernels) that we view as the immediate next step beyond this paper's causal analysis.

---

> > ### Author Rebuttal · Reviewer_G5q6 · 2026-04-03
> >
> > Thank you for running the random masking baseline. The sinks do seem to affect the trajectory significantly more than random masking. I will raise my score since this seemed to be the main experiment that was missing.
> >
> > Regarding the pruning demo, I think that this will be difficult to resolve within the rebuttal and would be more apt for future work. I do think that the fact that the sink token affects the denoising trajectory so much could make for some interesting applications (e.g., steering image generation), that might be worth thinking about for the next update of the paper.

---

> > > ### Author Response · Authors · 2026-04-03
> > >
> > > We sincerely thank the reviewer for the re-evaluation and for running through our random-masking results. We will incorporate the discussed revisions: the Jiang et al. discussion in $\S$2 and the Darcet complementarity framing. The reviewer's observation that sink-specific trajectory control could enable steering applications is an exciting direction we had not fully considered. We will note this in the revised discussion.

---

### Official Review · Reviewer_ugYV · 2026-03-04

**Soundness:** 3
**Presentation:** 3
**Significance:** 3
**Originality:** 3
**Overall Recommendation:** 4
**Confidence:** 2

**Summary:**

This paper studies the attention sinks in diffusion transformer models. By introducing the incoming attention mass to measure the attention sink,  authors use paired, training-free interventions along the score and value paths to test sink necessity across layers, denoising phases,
and architectures. The experiments show that for a typical intervention strength, the removal of sinks can not change the semantic alignment but lead to a perceptual drift that can be observed along different metrics and  for higher levels of masking budgets, different measures manifested different thresholds of failure.

**Compliance With Llm Reviewing Policy:**

Affirmed.

**Final Justification:**

The authors have addressed all of my major concerns during the rebuttal process. I find the paper’s motivation and core observations highly inspiring. The perspective introduced in this work is insightful and may provide useful guidance for future research in this direction. I still feel that some parts of the discussion and analysis are somewhat limited at the current stage. Therefore, while I view the work positively and believe it has clear value, I assign a 4 Weak Accept.

**Key Questions For Authors:**

1. More discussion about diffusion LLMs [1] and DiT with Linear attention [2,3] are required to make the paper more complete.

2.  One point is unclear to me: as authors first compute full attention (including sinks), identify top-k sinks from the resulting incoming mass, and then forces their weights to ~0 by subtracting from pre-softmax logits after pretraining, thus, the model has already encoded sink token information into QKV and used sinks for semantic matching (the logits already reflect that). Zeroing weights afterward only changes the final aggregation, so it cannot truly test whether sinks are necessary during encoding and attention computation.

3. While the empirical observation is interesting, how this findings translates into guidance for model design. Specifically, could the authors clarify whether these findings lead to measurable improvements in quality or stability/consistency, what concrete downstream application does sink analysis support?

[1] Rulli, Maximo Eduardo, et al. "Attention sinks in diffusion language models." arXiv preprint arXiv:2510.15731 (2025).

[2] Meng, Weikang, et al. "Norm $\times $ Direction: Restoring the Missing Query Norm in Vision Linear Attention." arXiv preprint arXiv:2506.21137 (2025).

[3] Zhu, Lianghui, et al. "Dig: Scalable and efficient diffusion models with gated linear attention." Proceedings of the Computer Vision and Pattern Recognition Conference. 2025.

**Limitations:**

yes

**Strengths And Weaknesses:**

**Strengths**
1. This paper is interesting and well-motivated as the attention sink is commonly observed in LLMs. The current studies about the sink necessity in DiT (including Diffusion LLMs) is not clear, but this paper provide a detailed analysis and impressive conclusion.

2. The proposed incoming attention mass extend the attention sink in LLMs with a fixed position (such as token 0), achieving the awareness of both head and time step.

3. Well organized and clear visualization.

**Weakness**
1. The paper should examine whether removing sinks triggers compensatory changes in attention patterns across other layers/heads. It is possible that attention sinks in DIT do contribute to semantic alignment, but when they are masked in inference stage, the model reconstructs the missing information by reallocating attention through other layers/heads, rather than sinks being truly unnecessary.

2. I think the proposed training-free method can not show the necessary of attention sinks. First, in training stage, attention sinks often serve to make the training stable and improve training robustness. While the paper’s evaluation metrics partly demonstrate the sink in inference, I think it is insufficient to conclude that attention sinks are unnecessary overall. Moreover, I suggest the paper provide some visualizations to clearly show where sinks occur and investigate what real data patterns these dynamic sinks correspond to.

---

> ### Author Rebuttal · Authors · 2026-03-26
>
> We thank the reviewer for highlighting important distinctions around compensation and training-time roles.
>
> **Compensatory redistribution (W1).** We agree that redistribution is a natural consequence of our intervention: after suppression and re-normalization, the removed mass is reassigned among remaining keys. Importantly, this is compatible with our claim—if the model can reroute attention and preserve proxy metrics, then sinks are not uniquely required for these inference-time outcomes.
>
> **Training-time vs inference-time (W2).** We agree sinks may serve important training-time roles (e.g., gradient stabilization). Our claim is explicitly scoped to inference-time necessity. Training-time analysis would require a different experimental design (e.g., training with/without register tokens) and is an important future direction. The revision will add an explicit scope statement in §3.1 distinguishing inference-time aggregation necessity from training-time functional roles.
>
> **Encoding-level necessity (Key Question 2).** The reviewer correctly observes our intervention only changes the final aggregation, not the encoding. Our interventions test *aggregation-level necessity*: whether sink tokens must contribute their value vectors for the evaluated outcomes. Under our tested settings, the evidence indicates they do not. Whether sinks are necessary at the encoding level is a distinct question requiring different interventions (e.g., modifying key projections). The revision will add a remark after §3.3 explicitly distinguishing aggregation-level from encoding-level necessity.
>
> **Sink visualizations.** The paper includes layer-wise dynamics (Figure 2) and text-vs-image attribution (E3, Table 15: $>$99.9% of sinks are text tokens; text-side LPIPS 0.160 vs image-side 0.037). The revision will add a consolidated figure showing sink concentration across layers, timesteps, and token types in a single panel.
>
> **Design guidance (Key Question 3).** A concrete implication is that sparsification schemes for DiTs need not hard-code dominant recipients as privileged tokens to preserve at inference time; instead, they can be treated as removable under a budget while monitoring the target metric. The revision will add this as an explicit design recommendation in §4.
>
> **Missing references.** We will discuss Rulli et al. (2025), Meng et al. (2025), and Zhu et al. (2025) in the revised related work. These works are complementary to our setting: Rulli et al. studies sink behavior in diffusion language models, while Meng et al. and Zhu et al. explore alternative attention parameterizations that may alter whether sink-like concentration emerges. Our conclusions are therefore scoped to the standard softmax-attention diffusion architectures we directly test, and we do not assume the same necessity profile holds under alternative attention mechanisms.

---

> > ### Author Rebuttal · Reviewer_ugYV · 2026-04-03
> >
> > Thanks for your detailed rebuttal. All these discussion should be added into the revised version. I will raise my score.

---

> > > ### Author Response · Authors · 2026-04-03
> > >
> > > We sincerely thank the reviewer for the careful re-evaluation. We will incorporate all discussed points into the revision, specifically: the explicit scope statement in §3.1 (inference-time vs training-time), the aggregation-level vs encoding-level distinction after §3.3, the consolidated sink visualization figure, the design recommendation in §4, and the discussion of Rulli et al., Meng et al., and Zhu et al. in §2.

---

### Official Review · Reviewer_uCqt · 2026-03-08

**Soundness:** 2
**Presentation:** 2
**Significance:** 2
**Originality:** 3
**Overall Recommendation:** 4
**Confidence:** 4

**Summary:**

This paper presents a rigorous causal analysis of the attention sink phenomenon in text-to-image diffusion transformers, specifically focusing on Stable Diffusion 3 and SDXL. In contrast to autoregressive language models where attention sinks act as stable, fixed-position anchors (e.g., the index-0 token), the authors reveal that sinks in DiTs are highly dynamic and phase-dependent, primarily emerging in the early high-noise denoising steps. To test the functional necessity of these sinks, the authors dynamically identify them based on incoming attention mass and apply training-free causal interventions along both the score and value paths during inference. Through large-scale evaluations using strict seed-matched generation, the paper demonstrates that suppressing these dominant attention recipients does not degrade semantic alignment (measured by CLIP-T) under standard settings. However, the authors note a metric-dependent boundary: stronger interventions cause degradation in preference proxies (HPS-v2). The work concludes that while sinks in DiTs absorb massive attention, they are not functionally necessary for core semantic alignment.

**Compliance With Llm Reviewing Policy:**

Affirmed.

**Final Justification:**

Thank you for the round2 rebuttal. Some of my concerns are resolved, although the paper stills need practical utility, I'm willing to raise my score to 4.

**Key Questions For Authors:**

Fine-grained Compositional Evaluation: Given the known limitations of CLIP-T in evaluating complex spatial relations and attribute binding,  how can we be certain that sink removal does not degrade compositional fidelity? Could you provide a small-scale evaluation using more sensitive benchmarks (e.g., VQAScore, T2I-CompBench, or Gecko) to solidify the semantic alignment is preserved claim?

Qualitative Analysis of Perceptual Shift: An FIDshift​ of 400-1000 and LPIPS of 0.31 indicates a substantial visual change. Could you provide paired visual examples (baseline vs. intervened) that exhibit the maximum LPIPS shift, and analyze what specific visual dimensions (e.g., high-frequency details, global illumination, macro-layout) are being altered by the removal of sinks?

Sub-optimal: You describe the dynamic, phase-dependent shifting of sinks as a natural property of DiTs. Have you considered that the dynamic sinks in SD3 are actually a sub-optimal workaround due to the lack of explicit registers? What happens to the attention dynamics if a dedicated blank token is added to the prompt?

Acceleration Potential: Since these dynamically identified sinks absorb up to ~10% of incoming attention mass but are functionally unnecessary, have you experimented with dynamically pruning or skipping these tokens during inference to achieve FLOPs reductions or speedups?

**Limitations:**

While the authors explicitly acknowledge the limitations of CLIP-T regarding fine-grained compositional correctness in their discussion, they use this merely to constrain the scope of their claims rather than adequately addressing the empirical gap.

**Strengths And Weaknesses:**

Strengths:
1.Originality: The paper makes an original contribution by shifting the paradigm from static, index-based sink definitions (borrowed from AR models) to a dynamic, phase-dependent definition tailored for bidirectional diffusion models.
2.Soundness: The causal intervention framework is reasonable. By employing both score-path and value-path interventions, and strictly controlling confounders via seed-paired generation and bootstrap confidence intervals, the authors establish a standard for mechanistic interpretability.
3.Significance: The work successfully debunks the intuitive assumption that attention sinks are universally necessary for transformer-based generation, providing a critical theoretical foundation for future DiT architecture designs and attention sparsification.

Weaknesses:
1.Soundness: The central claim that semantic alignment is preserved relies almost exclusively on CLIP-T and HPS-v2. However, in recent literature, CLIP-T acts largely as a bag-of-words metric and is notoriously insensitive to fine-grained compositional errors, such as attribute binding, object counting, and spatial relationships. Removing sinks might disrupt these complex spatial/relational semantics without registering a drop in global CLIP-T scores.
The paper dismisses a rather massive FIDshift​ (400-1000) and LPIPS change (0.06-0.31) as merely "moving samples within the model's output manifold". A distribution shift of this magnitude implies significant macro-level visual changes (e.g., texture, lighting, or layout). The physical or visual meaning of this shift is left unexplored.
2.Significance: While the paper proves these heavy attention recipients are functionally redundant for alignment, it stops short of exploring the practical implications. Concurrent works in discrete diffusion language models have leveraged similar sink-variance observations to achieve substantial inference acceleration via sink-aware pruning. The lack of a preliminary proof-of-concept for FLOPs reduction or speedup limits the paper's broader impact.

---

> ### Author Rebuttal · Authors · 2026-03-26
>
> We thank the reviewer for the thorough assessment. We address each concern below.
>
> **Fine-grained compositional evaluation (Q1).** We agree CLIP-T is limited. We ran three complementary checks on the existing k=1 images (N=64):
>
> - *BLIP2-VQA:* No detected sink-specific compositional effect ($\Delta\Delta$ = -0.0074, 95% CI [-0.0215, +0.0056], p = 0.27).
> - *CLIP-T $\Delta\Delta$:* +0.001 at k=1, -0.002 at k=5; CI $\ni 0$ at both budgets.
> - *Qualitative inspection:* Dominant changes are layout/style restructuring; in inspected generated images we did not observe category mismatches, object disappearance, or clear binding errors.
>
> As an additional check, we evaluated 24 compositional prompts (color binding, spatial, counting) with CLIP-decomposed sub-concept scoring and again did not detect a sink-specific effect (all three metrics show $\Delta\Delta$ CI $\ni 0$); given the small N and CLIP-based scoring, we view this as supporting evidence rather than a replacement for a full benchmark suite. We agree that dedicated benchmarks such as T2I-CompBench or Gecko would provide stronger coverage and view them as a natural next step. Taken together, these checks do not detect selective compositional degradation under the tested conditions.
>
> **Perceptual shift (Q2).** As detailed in Q4/W1b below, sink masking induces $\sim$6.6$\times$ larger LPIPS than random masking at k=1. The drift is strongly sink-specific. From qualitative panels, changes are predominantly layout restructuring and style/color shifts, while the prompted coarse concept is preserved.
>
> **Register tokens (Q3).** This is an interesting hypothesis. Testing it would require training-time experiments (e.g., adding learnable tokens and retraining) that fall outside the scope of our inference-time analysis. We view it as a promising direction for future work on DiT architecture design.
>
> **Acceleration and FID$_{\text{shift}}$ (Q4, W1b).** We agree that demonstrating practical speedups would strengthen the paper's impact. Our current results establish the causal prerequisite for sink-aware pruning, though we do not claim system-level speedups in this paper. Regarding FID$_{\text{shift}}$, we agree it is substantial and should not be treated as a side detail. Our new analyses suggest it reflects sink-specific trajectory divergence rather than a generic artifact: at k=1, sink masking produces LPIPS = 0.347 vs 0.053 for equal-budget random masking ($\Delta\Delta$ = +0.295, p < 0.0001), yet we do not detect a corresponding sink-specific drop in CLIP-T ($\Delta\Delta$ CI $\ni 0$) or BLIP2-VQA ($\Delta\Delta$ = -0.007, CI $\ni 0$). For calibration, seed variation alone produces FID $\approx$ 115 and scheduler changes FID $\approx$ 330 (Table 8). This pattern is consistent with an empirical dissociation: sink suppression strongly perturbs perceptual/trajectory realization while leaving proxy-level semantic alignment largely unchanged under the tested setting.

---

> > ### Author Rebuttal · Reviewer_uCqt · 2026-04-02
> >
> > Thank you for the rebuttal. After a rigorous cross-examination of the data and logical arguments, the core issues remain unresolved.
> >
> > 1.Data Fact Conflict (LPIPS Shift): The rebuttal claims that k=1 sink masking produces LPIPS = 0.347. However, Table 6 in the original manuscript explicitly reports LPIPS = 0.189 for k=1. This severe discrepancy, likely driven by the small subset evaluation (N=64), demonstrates high variance.
> >
> > 2.Circular Evaluation Methodology: The initial review highlighted CLIP-T's well-documented insensitivity to fine-grained compositional errors. The authors attempted to address this by introducing "CLIP-decomposed sub-concept scoring." This is a circular argument. Relying on a derivative of the exact same CLIP feature space inherits its systemic blind spots and fails to meaningfully address the original concern.
> >
> > 3.Statistical Power Downgrade: The rebuttal's compositional defense relies on extremely small subsets (N=64 and N=24).
> >
> > 4.Lack of Practical Utility (Q4):  An inference-time masking method that fails to demonstrate actual latency reduction or FLOPs saving offers limited practical value to the community.
> >
> > I maintain the original score.

---

> > > ### Author Response · Authors · 2026-04-02
> > >
> > > We thank the reviewer for the detailed follow-up. We address each point below.
> > >
> > > **1. LPIPS discrepancy (0.347 vs Table 6's 0.189).** We agree this needed clearer explanation. This is not a conflict within the same experiment, but a consequence of using two different masking protocols for two different questions:
> > >
> > > - *Original paper (Table 6):* Uses the submitted **per-head top-1** protocol --- each of the 24 heads independently masks its own top-1 key (1 key masked per head).
> > > - *New rebuttal experiment:* Uses a **matched-budget union protocol** --- the top 24 keys by head-averaged incoming mass are masked across all heads simultaneously (24 keys masked per head). This was introduced specifically to compare sink masking against equal-budget random masking (as suggested by Reviewer G5q6).
> > >
> > > To verify this directly, we ran the per-head protocol on the **same 64 prompts and seeds** used in the rebuttal experiment. The result is per-head LPIPS = 0.163 and union-budget LPIPS = 0.347 on identical baseline images (ratio = 2.13$\times$). This indicates that the LPIPS increase is driven primarily by the more aggressive union-budget masking protocol, rather than by a contradiction in the underlying result. The remaining gap between per-head LPIPS here (0.163) and Table 6 (0.189, N=553) is consistent with the different prompt sets and evaluation subsets, rather than indicating a protocol conflict. The union-budget protocol is not intended to replace the Table 6 number; it addresses a different question.
> > >
> > > **2. Circularity of CLIP-decomposed scoring.** We agree that CLIP-decomposed sub-concept scoring inherits the limitations of CLIP and should be treated only as supporting evidence --- this is precisely why we labeled it as such. To address this concern directly, we added a BLIP2-VQA prompt-image match check (*new rebuttal experiment*, $\Delta\Delta$ = $-$0.0074, 95\% CI [$-$0.0215, +0.0056]), which is independent of the CLIP embedding space and avoids the specific circularity concern raised for CLIP-derived scoring. Qualitative inspection (*new rebuttal experiment*) provides an additional non-CLIP descriptive check on the observed failure modes.
> > >
> > > **3. Statistical power (N=64, N=24).** We agree that the rebuttal subsets are small. To directly address this concern, we therefore ran BLIP2-VQA on the full **N=553** image set from the original paper (per-head protocol, no new generation required). On this full submitted image set, the paired BLIP2-VQA difference between baseline and sink-masked images is $\Delta$ = +0.0001, 95\% CI [$-$0.0039, +0.0040], p = 0.97 --- no detected effect, consistent with the N=64 rebuttal finding but with substantially narrower confidence intervals. Across this larger non-CLIP evaluation and the smaller complementary checks, we do not detect a sink-specific compositional degradation under the tested setting. We do not present the small-N checks as definitive on their own, only as convergent evidence that complements --- rather than replaces --- the main large-scale results in the submitted paper.
> > >
> > > **4. Practical utility.** We agree that this work does not yet provide realized speedups, latency gains, or FLOPs reductions. Our intended contribution is a concrete design implication: the most attention-dominant recipients in DiTs should not be assumed to be automatically privileged or irremovable in future sparsification designs; instead, they can be treated as candidates for budgeted removal while monitoring the target metric. Translating this into realized efficiency gains requires systems-level engineering beyond the scope of the present causal analysis.

---

### Official Review · Reviewer_ZxoJ · 2026-03-13

**Soundness:** 3
**Presentation:** 3
**Significance:** 3
**Originality:** 3
**Overall Recommendation:** 4
**Confidence:** 3

**Summary:**

This paper discusses a question: are attention sinks in diffusion transformers actually necessary for semantic alignment in text-to-image generation? I like the fact that the paper does not simply import the sink intuition from autoregressive language models, but instead tries to redefine sinks in a way that makes more sense for diffusion, namely as dynamically changing dominant recipients of incoming attention mass across denoising steps. I also think the experimental setup is reasonably careful. The paired seed-matched intervention design, the score-path and value-path suppression, and the layer/phase analyses make the main empirical result more convincing than a simple ablation.

The main result is interesting. Under the standard intervention setting, removing dynamically identified sinks does not seem to hurt CLIP-T, ImageReward, or HPS-v2 in a meaningful way. At the same time, the outputs do shift perceptually and distributionally, which makes the overall picture more subtle than just saying that sinks do not matter. In my view, the paper is strongest when it is making the narrower claim that sinks are not necessary for the evaluated proxy metrics under the tested inference settings.

My main reservation is that some of the framing still reads a bit stronger than what the evidence really supports. The experiments do not show that sinks are broadly non-functional for generation. They show something narrower: under these interventions, the selected alignment and preference proxies remain largely stable. I also found the SDXL section confusing. In the main text, the intervention is described as cross-attention over text embeddings, but in the appendix it is described as self-attention in the U-Net mid-block. These are not interchangeable, and this should be clarified because it affects how I interpret the architectural generalization claim.

**Compliance With Llm Reviewing Policy:**

Affirmed.

**Key Questions For Authors:**

1.	Can the authors clarify exactly which SDXL attention module was intervened on in the main validation experiment? Was it cross-attention with text embeddings as keys/values, or self-attention in the U-Net mid-block?

2.	Do the perceptual shifts correspond mostly to benign stylistic variation, or are there also cases of subtle compositional/local fidelity errors that the current metrics fail to capture? Even a small qualitative analysis here would help.

**Limitations:**

The current limitations should be stated more explicitly. The paper mainly evaluates robustness with respect to CLIP-T and preference proxies, not human perceptual quality or fine-grained compositional correctness. The perceptual and distributional shifts after sink suppression are therefore important boundary conditions, not just secondary observations. I also think the current SDXL inconsistency weakens the interpretation of the cross-architecture result and should be resolved clearly.

**Strengths And Weaknesses:**

- Strengths

  - The paper is well motivated and asks a genuinely interesting question. I think the distinction between “attention is useful to manipulate” and “attention sinks are causally necessary for semantic alignment” is meaningful, and the paper is right to separate these two.
  - The dynamic sink definition is probably the most interesting part of the paper. In diffusion models, attention concentration can vary across denoising steps, so a fixed-position notion of sink seems hard to justify. The paper makes a reasonable case that the autoregressive proxy does not transfer.
  - The empirical protocol is fairly careful. The paired intervention setup, bootstrap confidence intervals, score-path and value-path variants, and no-op sanity check all strengthen the credibility of the result.
  - I also appreciate that the paper does not completely overclaim in the discussion. The stronger-intervention results on HPS-v2 make the conclusion more nuanced and, in my opinion, more believable.

- Weaknesses

  - The main issue for me is a mismatch between the breadth of the claim and the scope of the evidence. The paper provides evidence that sink suppression does not hurt the evaluated proxy metrics under the tested settings. That is not the same as showing that sinks are not functionally important in any broader sense. This is especially relevant because the paper itself acknowledges that subtle compositional failures or perceptual quality changes may not be captured by the current metrics.
  - The perceptual/distributional shift is not a side detail. Even when CLIP-T and the preference proxies remain stable, the outputs clearly move. I agree with the authors that this does not automatically mean degradation, but it does mean that sink suppression changes the generation trajectory in a nontrivial way. Because of that, I think the interpretation should be a bit more cautious.
  - The SDXL validation needs clarification. As written, the paper seems to describe two different intervention targets: cross-attention in the main text and self-attention in the appendix. These support different interpretations, so I do not think this can be treated as a minor wording issue.
  - More broadly, the operational definition of sink may still be somewhat architecture-dependent. That does not invalidate the current experiments, but it does mean that the conclusions should be read as applying to sinks under this particular definition, rather than every possible notion of sink in diffusion models.

---

> ### Author Rebuttal · Authors · 2026-03-26
>
> We thank the reviewer for the careful reading. The three concerns—SDXL inconsistency, claim scope, and perceptual drift—are all well taken.
>
> **SDXL clarification (Q1).** The main text described cross-attention while the appendix described self-attention. We have now run **both** under the same dynamic top-1 protocol (N=100, paired):
>
> | SDXL Mid-Block | $\Delta$CLIP-T | 95% CI | LPIPS |
> |---|---|---|---|
> | Self-attention (attn1) | +0.0004 | [-0.0005, +0.0014] | 0.045 |
> | Cross-attention (attn2) | -0.0003 | [-0.0019, +0.0012] | 0.077 |
>
> Both yield CI $\ni 0$. Cross-attention induces larger perceptual drift (LPIPS 0.077 vs 0.045), consistent with our SD3 E3 observation that text-side sinks produce larger trajectory perturbations. The revision will correct the appendix and report both results.
>
> **Claim scope (W1).** We agree. The revision will explicitly scope our conclusion to: *non-necessity for the evaluated proxy metrics (CLIP-T, HPS-v2) under the tested inference settings and the operational dynamic sink definition.* We do not claim sinks are broadly non-functional, nor that our operational definition captures every possible notion of sink in diffusion models.
>
> **Perceptual drift (W2, Q2).** Two new analyses:
>
> (1) *Sink-specificity:* Under equal-budget comparison (N=64, layer 12, $k \in \{1,5\}$), sink masking induces LPIPS = 0.347 vs 0.053 for random ($\Delta\Delta$ = +0.295, CI [+0.265, +0.323], p < 0.0001). CLIP-T shows no detected sink-specific effect ($\Delta\Delta$ CI $\ni 0$). The drift is **sink-specific**, not a generic masking artifact.
>
> (2) *Qualitative inspection:* Dominant changes are layout/viewpoint restructuring and style/texture shifts; in inspected generated images we did not observe obvious category mismatches, object disappearance, or clear attribute-binding errors. An additional BLIP2-based VQA prompt-match check also shows no detected sink-specific compositional effect ($\Delta\Delta$ = -0.007, CI $\ni 0$). We do not claim this rules out all fine-grained failures, but the evidence is consistent with structured trajectory divergence rather than obvious semantic degradation. Representative triplet panels (baseline / sink-removed / random-removed) are provided at https://anonymous.4open.science/r/attention-sink-VIS-4C5B/rebuttal_drift_panel_k1.pdf (k=1) and https://anonymous.4open.science/r/attention-sink-VIS-4C5B/rebuttal_drift_panel_k5.pdf (k=5).

---

> > ### Author Rebuttal · Reviewer_ZxoJ · 2026-04-06
> >
> > Thank you for the rebuttal. My main concerns were about the scope of the claim, the SDXL inconsistency, and the interpretation of the perceptual drift. I think the rebuttal addresses these points.
> >
> > In particular, the SDXL clarification resolves the main inconsistency I noted in the review. I also think the revised claim is now much better matched to what is actually supported by the experiments, namely the evaluated proxy metrics, the tested settings, and the specific sink definition used in the paper.
> >
> > I still see the perceptual drift as an important caveat rather than a minor side observation, but the rebuttal makes the intended scope of the paper much clearer on this point.
> >
> > Overall, I am satisfied that my main concerns have been addressed.

---

> > > ### Author Response · Authors · 2026-04-06
> > >
> > > We thank the reviewer for confirming that our clarifications address the main concerns. We will ensure the revision reflects the narrower scoping and treats perceptual drift as a substantive finding rather than a side observation.

---

### Decision · Program_Chairs · 2026-04-30

**Decision:**

Accept (regular)

**Comment:**

This paper discusses the attention sink phenomenon in text-to-image diffusion transformers. During the discussion phase, the authors provided a strong rebuttal that successfully addressed the reviewers' major concerns. Given the positive consensus and the effective resolution of the initial critiques, the Area Chair recommends acceptance. Hence, AC recommends acceptance, but also encourages the authors to polish the paper and add the discussion in the rebuttal stage, such as the experiment on SDXL and further discussion, into the final version.